# HYDROGEN: HYDROLOGICAL REPORT GENERATION WITH TWO-STAGE INSTRUCTION-TUNED MULTIMODAL MODELS, TEMPORAL PROMPTS, & KNOWLEDGE-GUIDED AGENTS

## ABSTRACT

Hydrological report generation is crucial for monitoring clouds, typhoons, rainfall, and water events, yet current multimodal models struggle with multi-image alignment and domain-specific knowledge integration. We present HydroGen, a domain-adaptive framework that overcomes these challenges through instruction tuning, temporal modeling, and knowledge-guided reasoning within a two-stage pipeline. First, we build HydroMM-Instruct, an instruction dataset that uses YOLOv8 for typhoon detection and shapefiles for region mapping, with reports standardized into cause–effect phrases. Second, we introduce a two-stage training pipeline with continual pre-training on hydrological data (radar maps, pressure charts, expert assessments) followed by supervised fine-tuning for report generation. Third, to enhance multi-image alignment, we introduce temporal prompt tokens that capture event sequences and progressions. Finally, we present GuideAC, an in-context agent that injects antecedent–consequence rules to improve reasoning. Evaluation on Thailand's weekly hydrological reports (2018–2025) shows that HydroGen substantially outperforms strong multimodal baselines, achieving a BERT-F1 of 84.56% (+46.96%) and a ROUGE-L of 67.78% (+61.48%).

## 1 INTRODUCTION

Understanding and explaining hydrological events—such as typhoons (Wang et al., 2016), precipitation (Adler et al., 2018), and flood development (Doocy et al., 2013)—play a vital role in monitoring natural phenomena, which directly affect water resources, disaster management, and environmental planning (Sene, 2010; Nemec, 2012). These studies often integrate satellite and radar observations, organize temporal monitoring sequences, and apply domain expertise to explain the links between atmospheric processes and hydrological responses (Fletcher, 1951; Cudennec et al., 2016; Yoo et al., 2020). Traditional hydrological event reporting primarily relies on human expertise (Pagano et al., 2016). To mitigate these limitations, machine learning and artificial intelligence have been progressively utilized in hydrological reporting, enhancing forecasting precision, computational efficiency, and response time to extreme occurrences (Costa et al., 2024). By leveraging multiple data sources and bypassing the complicated structure of conventional physical models (Antunes Meira & Xuan, 2024), AI delivers more prompt and accurate insights. It also supports data-informed analysis, making it a crucial tool for improving the quality and dependability of hydrological event reporting (Msigwa & Makinde, 2024; Marshall et al., 2025). However, contemporary automated techniques often inadequately incorporate both multimodal visual and textual attributes in the hydrological domain.

The rise of Multimodal Large Language Models (MLLMs), which are engineered to analyze and generate information across several modalities, including text, images, and videos, enabling the synthesis of visual and verbal input to produce fluent and applicable outputs, for example, GPT-4V (OpenAI, 2023), LLaVA (Li et al., 2024a), Qwen2.5-VL (Bai et al., 2025) and Gemma (Team et al., 2025). Such models have been successfully employed in several fields, including medical imaging (Xiao et al., 2024; Ye & Tang, 2025), chemistry (Liao et al., 2024; Li et al., 2025), and geological observation (Liu et al., 2024; Huang et al., 2025c; Xiao et al., 2025), demonstrating strong

capabilities in reasoning, information consolidation, and contextual understanding. In spite of these advancements, its utilization in the hydrological field remains predominantly unexplored. Weekly water reports present unique challenges: events must be linked to their locations (e.g., typhoons and affected areas), images aligned across multiple time points, and cause–effect relations captured (e.g., typhoons causing flooding), making report generation in this domain significantly more difficult than in others.

To mitigate these constraints, we present **HydroGen**, the debut multimodal framework for hydrological report creation. HydroGen employs advancements in preliminary processing, domain-adaptive training, temporal reasoning, and expert-guided adaptation to connect multimodal large language models with hydrological intelligence. In conclusion, our contributions comprise the following:

1. We integrate image preprocessing with YOLOv8 (Jocher et al., 2023) and shapefile mapping, allowing the system to identify typhoons and delineate their effects on specific areas. This approach improves regional precision and delineates explicit correlations between imaging data and impacted areas. Additionally, we normalize the report into a cause–effect writing style, reformulating hydrological reports into a cohesive narrative using cloud images and air pressure maps. This ensures stylistic consistency and facilitates integrated reasoning across modalities. The aim of this step is to generate a strong hydrological instruction dataset to train HydroGen, called **HydroMM-Instruct**, which consists of 431 textual reports with 5,590 images.

2. We propose a **two-stage training strategy** for HydroGen. Initially, we conduct hydrological continual pre-training that utilizes hydrology-specific resources such as radar data, pressure maps, textbooks, and expert reports to endow the model with domain-specific expertise. Subsequently, supervised fine-tuning is conducted on a multimodal hydrological dataset that includes satellite photos, air pressure maps, and reports. This methodology improves model comprehension related to creating water reports and ensures conformity with established reporting requirements.

3. We incorporate **temporal prompt tokens** between image inputs, delivering explicit ordering signals that allow MLLMs to produce coherent narratives which reflect the chronological order of hydrological events.

4. We construct an in-context learning agent that synthesizes hydrology-related information into antecedent-consequence rules entitled **GuideAC**. This bolsters HydroGen's capacity to identify causal linkages and improves the clarity of generated reports.

We evaluate HydroGen on weekly hydrological reports from Thailand's Hydro-Informatics Institute (HII), spanning 2018–2025 and including diverse inputs such as air pressure maps, satellite imagery, and expert-authored reports. HydroGen generates reports that accurately link clouds and typhoons to rainfall and related water events while maintaining domain-specific clarity. Empirically, it outperforms strong off-the-shelf multimodal baselines, achieving a BERT-F1 of 84.56% (+46.96%) for semantic accuracy and a ROUGE-L of 67.78% (+61.48%) for syntactic fidelity.

## 2 RELATED WORKS

**AI for Hydrology, Meteorology, and Disaster Forecasting.** Previous AI strategies for scientific applications have primarily focused on numerical models (Hussain et al., 2020; Deng et al., 2024), such as FourCastNet (Pathak et al., 2022) and GraphCast (Lam et al., 2023) are used for precipitation forecasting, or on image-based techniques, such as TCICENet (Zhang et al., 2021) and DeepTyphoon (Lu et al., 2022) adopting LSTM and CNN for typhoon recognition from satellite data. While useful for particular tasks, these technologies lack the capability for textual analysis and the development of natural language outputs, which are essential for creating expert-level hydrological reports.

**Domain-Specific MLLMs.** Universal MLLMs have demonstrated a robust capacity for blending text and visual modalities, rendering them proficient for several open-domain activities, including question answering, captioning, and reasoning. Several research projects have adapted MLLMs for scientific purposes, building on these foundations. For instance, PEACE (Huang et al., 2025b) facilitates multimodal agents in geological map interpretation, TEOChat (Irvin et al., 2025) represents

an earth observation chatbot for the analysis of remote sensing data, and CLLMate (Li et al., 2024b) enhances climatic event forecasting with multimodal inputs. These research highlight the versatility of LLaVA-based architectures (Liu et al., 2023) in specialized scientific domains, demonstrating that MLLMs can be customized to align with domain knowledge and task specifications. Nonetheless, despite these advancements, hydrological comprehension and documentation remain predominantly unexamined. In contrast to geology or climate forecasting, hydrology necessitates the clarification of the causal interactions among clouds, typhoons, precipitation, and their effects on water systems.

**Knowledge Guidance In-Context Learning.** In-context learning enables Large Language Models (LLMs) to adjust to novel tasks by the provision of examples or prompts, without necessitating updates to the model parameters (Dong et al., 2022), frequently employing methods like few-shot prompting (Brown et al., 2020), ReAct prompting (Yao et al., 2023) or guided rules (Pang et al., 2023). Previous studies, such as GuideNER (Huang et al., 2025a), utilize named-entity-based rules to direct LLM results inside textual contexts. Nonetheless, these methodologies are not directly applicable to MLLMs and are constrained in the development of hydrological reports, which necessitate both visual and textual reasoning. Building on this line of research, we will extend the knowledge-guidance strategy from GuideNER to address the unique challenges of hydrological report generation.

## 3 METHODS

### 3.1 OVERVIEW OF HYDROGEN FRAMEWORK

HydroGen starts by preprocessing multimodal hydrological data, turning images and expert observations into a consistent cause-and-effect model, as shown in Figure 1. Following the LLaVA design (Liu et al., 2023), these inputs are put together into a vision-language pipeline that combines a vision encoder, projector, and language backbone into one representation.

Beyond this backbone, HydroGen incorporates several innovations tailored for hydrological reasoning: a domain-specific instruction dataset namely HydroMM-Instruct, a two-stage training strategy for continual adaptation, temporal prompt tokens for multi-image alignment, and a knowledge-guided agent called GuideAC for causal reasoning. These components, detailed in the following sections, collectively strengthen the model's ability to generate expert-like hydrological reports.

In practice, we first prepare satellite images and air pressure maps using an image preprocessing module for typhoon recognition and regional mapping, while metadata is incorporated into the instruction with consolidated ground truth produced by the style transfer module. Subsequently, train the MLLM to integrate these inputs with temporal prompts that synchronize sequential images, allowing the model to understand its progression of events. During inference, GuideAC enhances causal reasoning. Lastly, the predicted reports are assessed against ground truth reports.

### 3.2 HYDROMM-INSTRUCT DATASET CONSTRUCTION

#### 3.2.1 DATASET ACQUISITION ($\mathbb{D}_{\text{RAW}}$)

We obtain data from the Hydroinformatics Data Center Platform[1] that the Thailand Hydro-Informatics Institute (HII) operates. This platform provides open-access weekly reports on the state of the hydrological situation. The record includes information from various sources, such as satellite images, weather stations on the ground, and regional monitoring platforms. Its weekly updates are aimed to keep the public, policymakers, and disaster response teams up-to-date on hydrological hazards that are still unfolding. Usually, each report contains a number of distinctive events, such as typhoons, the quantity of rain, how saturated the soil is, the capacity of the dams, and the degree of the air pressure, as well as how numerous floods have occurred in Thailand.

For this study, we specifically extract two sections as shown in Figure 1: (i) satellite images and (ii) air pressure maps. These contain both visual observations of cloud movement and storm systems,

---

[1] https://www.thaiwater.net

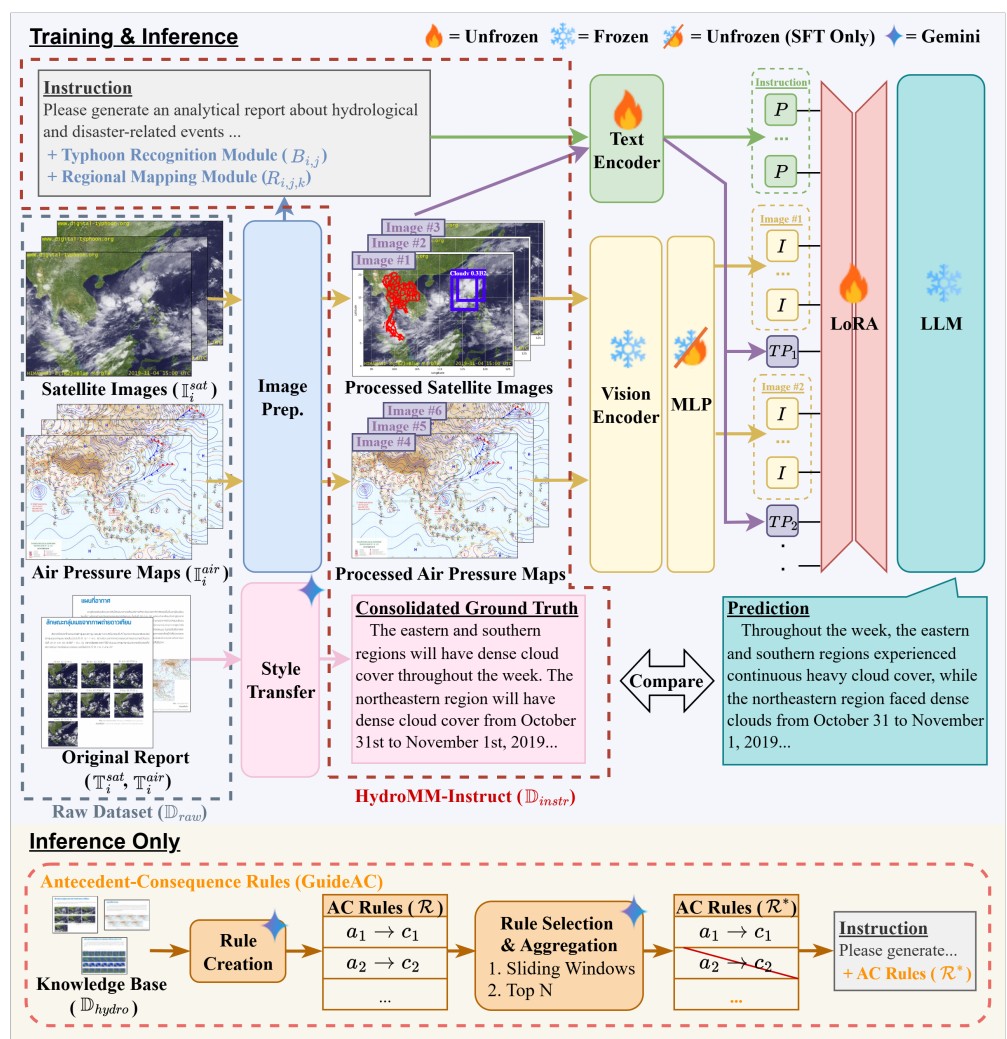

Figure 1: Overall architecture of HydroGen. The instruction and results were originally in Thai and have been translated into English.

as well as textual expert summaries describing their evolution. This selection ensures a consistent multimodal pairing between visual evidence and explanatory narrative, which is critical for instruction-style dataset construction.

The temporal coverage of our dataset spans from January 2018 to October 2025, yielding $N$ weekly reports. Each report provides a textual component $\mathbb{T}_i$ and an associated image set $\mathbb{I}_i$, forming the raw dataset:

$$\mathbb{D}_{\text{raw}} = \{(\mathbb{T}_i, \mathbb{I}_i)\}_{i=1}^{N}, \tag{1}$$

Where each element corresponds to one weekly report containing both narrative and visual components. In each image set $\mathbb{I}_i$, we include up to seven satellite images and up to seven air pressure maps. Formally, we define:

$$\mathbb{I}_i = \{\boldsymbol{I}_{i,j}\}_{j=1}^{M_i}, \quad M_i = |\mathbb{I}_i^{\text{sat}}| + |\mathbb{I}_i^{\text{air}}|, \ |\mathbb{I}_i^{\text{sat}}| \leq 7, \ |\mathbb{I}_i^{\text{air}}| \leq 7, \tag{2}$$

Where each $\boldsymbol{I}_{i,j}$ is either a satellite image $\boldsymbol{I}_{i,j} \in \mathbb{I}_i^{\text{sat}}$ or an air pressure map $\boldsymbol{I}_{i,j} \in \mathbb{I}_i^{\text{air}}$.

### 3.2.2 Typhoon Recognition and Regional Mapping ($B_{i,j}$, $R_{i,j,k}$)

We process the image set $\mathbb{I}_i$ in each report to identify typhoons and cloud systems. As shown in the blue *Image Prep.* box in Figure 1, For each image $\boldsymbol{I}_{i,j} \in \mathbb{I}_i$, we use a YOLOv8 (Jocher et al., 2023)

object detector fine-tuned on typhoon and cloud images provided by the HII database. The detector outputs bounding boxes:

$$\boldsymbol{B}_{i,j} = f_\theta^{\text{YOLO}}(\boldsymbol{I}_{i,j}), \tag{3}$$

Where $\boldsymbol{B}_{i,j}$ holds the box coordinates and confidence scores. Then for each satellite image $\boldsymbol{I}_{i,j} \in \mathbb{I}_i^{\text{sat}}$ is mapped to its geographical region using official shapefile data, which contains the latitude–longitude boundaries and area polygons of all Thailand provinces. This gives a region label $R_{i,j,k} = \phi(\boldsymbol{I}_{i,j})$. The multimodal representation of report $i$ is defined as:

$$\boldsymbol{X}_i = \{(\boldsymbol{I}_{i,j}, \boldsymbol{B}_{i,j}, R_{i,j,k})\}, \tag{4}$$

### 3.2.3 CAUSE–EFFECT STANDARDIZATION ($\mathbb{Y}_i$)

The textual reports $\mathbb{T}_i$ vary in style across years, with separate narratives for satellite observations and air pressure analyses. As shown in the pink *Style Transfer* box in Figure 1, we first decompose each report into its satellite-based component $\mathbb{T}_i^{\text{sat}}$ and air pressure-based component $\mathbb{T}_i^{\text{air}}$. We then apply Gemini 2.5 (Comanici et al., 2025) to rewrite them into a unified cause–effect form:

$$\mathbb{Y}_i = g_\theta(\mathbb{T}_i^{\text{sat}}, \mathbb{T}_i^{\text{air}}), \tag{5}$$

Here, $\mathbb{Y}_i$ represents a standardized explanation where meteorological *causes* (e.g., dense cloud coverage, high air pressure, rapid wind circulation) are explicitly linked to their hydrological *effects* (e.g., rainfall, flooding, or typhoon formation). Additional rewriting rules—such as normalization of terminology and template phrasing—are implemented through curated prompts, as detailed in the Appendix B.1. Hydrological experts review a subset of these standardized outputs to validate their accuracy and domain consistency.

### 3.2.4 INSTRUCTION-TUNING PREPARATION ($\mathbb{D}_{\text{INSTR}}$)

Finally, each report is transformed into an instruction-following example by combining multimodal inputs with standardized outputs. This process, highlighted by the red dashed *HydroMM-Instruct* box in Figure 1, produces the instruction dataset:

$$\mathbb{D}_{\text{instr}} = \{(\boldsymbol{X}_i, P_i, \mathbb{Y}_i)\}_{i=1}^N, \tag{6}$$

Where:

- $\boldsymbol{X}_i$: multimodal input (satellite images, bounding boxes, and mapped regions),
- $P_i$: an instruction-style prompt (the prompt design are provided in Appendix B.2),
- $\mathbb{Y}_i$: expert-validated cause–effect narrative derived from $\mathbb{T}_i$.

### 3.3 TWO-STAGE HYDROLOGICAL TRAINING STRATEGY

Generic multimodal large language models (MLLMs) are not designed with hydrological expertise. Directly fine-tuning them on a limited hydrology dataset can lead to poor generalization and factual errors. To address this, we adopt a two-stage training strategy as illustrated by the *Unfrozen* components in Figure 1. First, we adapt the base MLLM to hydrological knowledge using continual pre-training (CPT). Then, we refine its ability to produce expert-style reports with supervised fine-tuning (SFT) on HydroMM-Instruct. This gradual adaptation allows the model to acquire domain-specific concepts before aligning to hydrological reasoning and reporting tasks.

### 3.3.1 CONTINUAL PRE-TRAINING ( 🔥 )

In the first stage, the model is continually pre-trained using hydrological knowledge base including textbooks, expert-authored reports, and textual summaries of radar and pressure maps. Each text sample is denoted $\mathbb{C}_j \in \mathbb{D}_{\text{hydro}}$. The training objective is causal language modeling (CLM), where the model predicts masked tokens from text:

$$\min_\theta \ \mathbb{E}_{\mathbb{C}_j \sim \mathbb{D}_{\text{hydro}}} \ \mathcal{L}_{\text{CLM}}(f_\theta(\mathbb{C}_j)). \tag{7}$$

This stage equips the model with foundational hydrological knowledge, domain-specific terminology, while preserving general language understanding.

### 3.3.2 SUPERVISED FINE-TUNING (🎣, 🔥)

In the second stage, we fine-tune the model on the HydroMM-Instruct dataset introduced in Formula 6. Each training sample consists of multimodal inputs $\boldsymbol{X}_i$, an instruction prompt $P_i$, and the corresponding expert-standardized report $\mathbb{Y}_i$. The training objective is next token prediction over the standardized cause-effect narratives:

$$\min_\theta \; \mathbb{E}_{(\boldsymbol{X}_i, P_i, \mathbb{Y}_i) \sim \mathbb{D}_{\text{instr}}} \; \mathcal{L}_{\text{NLL}}(f_\theta(\boldsymbol{X}_i, P_i), \mathbb{Y}_i), \tag{8}$$

Where $\mathcal{L}_{\text{NLL}}$ denotes the negative log-likelihood of generating $\mathbb{Y}_i$ conditioned on $(\boldsymbol{X}_i, P_i)$. This aligns the model outputs with hydrological expert reasoning and reporting style.

### 3.4 TEMPORAL PROMPT TOKENS FOR MULTI-IMAGE ALIGNMENT ($TP_j$)

Hydrological reports often contain multiple sequential images, including satellite observations and air pressure maps. We introduce *temporal prompt tokens*, which indicate the position of each image in the sequence.

For each image in report $i$, we prepend a temporal prompt token, $TP_j$, where $j$ denotes the image order. In our dataset, we set the maximum number of temporal prompts to 14, corresponding to the maximum number of images per report (7 satellite + 7 air pressure maps). Reports with fewer images use only the subset of prompts required. The multimodal input with temporal prompts is then:

$$\tilde{\boldsymbol{X}}_i = \{(\boldsymbol{I}_{i,j}, \boldsymbol{B}_{i,j}, R_{i,j,k}, TP_j)\}_{j=1}^{M_i}, \quad M_i \le 14, \tag{9}$$

Where $(\boldsymbol{I}_{i,j}, \boldsymbol{B}_{i,j}, R_{i,j,k})$ is defined as in Formula 6. During supervised fine-tuning on HydroMM-Instruct, these temporal prompt tokens are passed to update the model parameters. This helps the model follow image sequences in report generation.

### 3.5 GUIDEAC: KNOWLEDGE-GUIDED CAUSAL REASONING ($\mathcal{R}^\star$)

Understanding hydrological events requires more than pattern recognition; it also depends on reasoning about cause and effect. Generic MLLMs often struggle to capture these implicit causal relations, such as "dense cloud coverage $\rightarrow$ rainfall" or "high-pressure system $\rightarrow$ typhoon weakening." To address this, we design *GuideAC*, a antecedent-consequence reasoning component that strengthens model outputs with external domain rules.

We first apply Gemini-2.5-Pro (Comanici et al., 2025) in an in-context learning setup to extract antecedent–consequence pairs from the hydrological corpus $\mathbb{D}_{\text{hydro}}$ (the detailed prompt is provided in Appendix B.3). Each extracted rule is written in the format:

$$\mathcal{R} = \{(a_\ell \rightarrow c_\ell)\}_{\ell=1}^L, \tag{10}$$

Where $a_\ell$ denotes a meteorological antecedent (e.g., strong wind circulation, dense clouds), and $c_\ell$ denotes the corresponding hydrological consequence (e.g., flooding, heavy rainfall).

Next, we aggregate the rules by grouping repetitive outputs with another Gemini-2.5-Pro agent. Then apply a Top-$N$ selection with a *sliding window* (as discussed in Appendix C.2). Ensuring the most frequent and stable rules are kept as the selected rule set, denoted $\mathcal{R}^\star$, which represents reliable causal patterns in hydrology.

Finally, during inference, $\mathcal{R}^\star$ is inserted into prompts as external context. This knowledge-guided prompting encourages the model to align its reasoning with expert-validated causal logic, improving both factual correctness and interpretability. The overall generation process in HydroGen is defined as:

$$\tilde{\mathbb{Y}}_i = f_\theta(\boldsymbol{X}_i, P_i, \mathcal{R}^\star), \tag{11}$$

Where the MLLM produces the hydrological report $\tilde{\mathbb{Y}}$ by integrating multimodal evidence $\boldsymbol{X}_i$, the instruction prompt $P_i$, and the chosen causal rules $\mathcal{R}^\star$.

Table 1: Rolling year-level cross-validation splits and dataset sizes for HydroMM-Instruct.

| Fold | Training Size | Validation Size | Test Size |
|---|---|---|---|
| Fold 1 (Train 2018–2020, Val 2021, Test 2022) | 177 | 59 | 61 |
| Fold 2 (Train 2018–2021, Val 2022, Test 2023) | 236 | 61 | 55 |
| Fold 3 (Train 2018–2022, Val 2023, Test 2024) | 297 | 55 | 37 |
| Fold 4 (Train 2018–2023, Val 2024, Test 2025) | 352 | 37 | 42 |
| **Total (Val/Test)** | – | **212** | **195** |

## 4 EXPERIMENTS AND RESULTS

### 4.1 DATASET

We evaluate HydroGen using the HydroMM-Instruct dataset presented in Section 3.2. The dataset consists of weekly hydrological reports from January 2018 to October 2025, covering multiple typhoon seasons, monsoon cycles, and regional flood events. Each report includes multimodal inputs $X_i$, such as satellite images, air-pressure maps, YOLO-based bounding boxes, region mappings, and temporal prompt tokens. Instruction prompts $P_i$ accompany each multimodal input, while the expert-standardized cause–effect narratives $\mathbb{Y}_i$ serve as the reference outputs.

To facilitate a comprehensive assessment that incorporates annual hydrological variability, we implement a rolling year-level cross-validation strategy. In each iteration of the analysis, the preceding years are utilized as the training dataset, the subsequent year is designated as the validation dataset, and the ensuing complete hydrological year is employed as the test dataset. The configuration and dataset sizes for all four folds are summarized in Table 1. Each test fold covers an entire hydrological cycle, which includes the monsoon onset (March to June), the monsoon peak (July to October), and the monsoon retreat (November to February). This evaluation strategy guarantees a comprehensive assessment of HydroGen across all seasonal periods, rather than limiting the examination to a narrow temporal window.

### 4.2 EVALUATION METRICS

To measure the quality of generated hydrological reports, we adopt both semantic and syntactic metrics. Semantic evaluation includes BERT (Zhang et al., 2019) and Sentence Similarity (SS) using PhayaThaiBERT (Sriwirote et al., 2023) to capture the meaning of Thai text. Syntactic evaluation includes BLEU (Papineni et al., 2002) and ROUGE-L (Lin, 2004) to quantify fluency and structural similarity to expert reports.

### 4.3 IMPLEMENTATION DETAILS

We use a partially frozen fine-tuning method, which is usual in LLaVA (Liu et al., 2023) and other MLLM training procedures. The vision encoder remains frozen to preserve pretrained visual representations, while the language backbone and MLP layers are unfrozen so they can be adapted to hydrological tasks. To facilitate parameter-efficient optimization, we implement LoRA (Hu et al., 2022) with a rank of 16, therefore reducing the number of trainable parameters to under 1% of the entire model. Training is conducted for 5 epochs through the Adam optimizer, with early stopping implemented after 9 steps. All experiments are performed on a singular NVIDIA A100 GPU with a batch size of 16, which strikes a good balance between memory limits and gradient stability.

**Scalability and Computational Efficiency.** HydroGen is engineered for computational efficiency and scalability. When the visual encoder is fixed and only the lightweight LoRA adapters are modified, the GPU memory demands are reasonable, with complete model training utilizing around 60 GB of VRAM on an A100. During inference, HydroGen utilizes less than 20 GB of memory and attains real-time decoding rates of 9–11 tokens/sec on an A100 GPU and 5–7 tokens/sec on a L4 GPU. The parameter-efficient components remain constant irrespective of dataset size, resulting in computational costs that scale mainly with data volume rather than model complexity. This enables

Table 2: Model performance comparison on semantic and syntactic metrics (%) including BERT-F1 (**B-F1**), Sentence Similarity (**SS**), BLEU score (**BLEU**) and ROGUE-L (**R-L**). The best results are highlighted.

| Model Name | Model Size | Semantic | | Syntactic | |
|---|---|---|---|---|---|
| | | B-F1↑ | SS↑ | BLEU↑ | R-L↑ |
| *Proprietary MLLMs* | | | | | |
| Gemini-2.5-Pro (Comanici et al., 2025) | - | 71.81 | 61.37 | 19.31 | 22.37 |
| GPT-4.1 (OpenAI, 2025) | - | 70.06 | 60.95 | 17.26 | 21.57 |
| *Open-source MLLMs* | | | | | |
| Gemma-3 (Team et al., 2025) | 4B | 66.05 | 48.01 | 6.00 | 13.92 |
| Typhoon2-Vision (Pipatanakul et al., 2024) | 7B | 37.60 | 34.73 | 0.41 | 6.30 |
| Qwen2.5-VL (Bai et al., 2025) | 7B | 51.90 | 46.22 | 7.67 | 16.29 |
| Llama-3.2-Vision (Dubey et al., 2024) | 11B | 34.41 | 22.84 | 0.59 | 4.96 |
| HydroGen-Typhoon2 (Ours) | 7B | **84.56** | **79.52** | **62.01** | **67.78** |

the framework to be utilized for larger hydrological datasets or weekly operational reporting without requiring architectural changes.

## 4.4 BASELINE MODELS

We evaluate HydroGen against both training baselines and evaluation-only baselines. For the ablation study and fine-grained comparison, we adopt two relatively small multimodal large language models (MLLMs) that support Thai, making them suitable for hydrological report generation tasks. In addition, we include stronger proprietary and open models as evaluation-only baselines to provide an upper bound reference.

- **Typhoon2-Qwen2-Vision-7B** (Pipatanakul et al., 2024). Part of the Typhoon 2 family, this model builds on Qwen2-VL (Wang et al., 2024), which integrates Naive Dynamic Resolution and Multimodal RoPE for efficient spatial–textual fusion. Typhoon2-Vision is further tuned on Thai data for tasks such as OCR and document understanding. We adopt the SCB10X checkpoint (`scb10x/typhoon2-qwen2vl-7b-vision-instruct`), instruction-tuned for both Thai and English.
- **Gemma-3-4B** (Team et al., 2025). From the Gemma series of lightweight instruction-tuned models, the 4B variant balances efficiency and performance. Gemma 3 improves long-context reasoning, alignment, and multilingual coverage. We adopt the checkpoint `google/gemma-3-4b-it`, which supports Thai, English, and other major languages.

To benchmark HydroGen against stronger systems, we also report results from (i) proprietary MLLMs involving **Gemini-2.5-Pro** (Comanici et al., 2025) and **GPT-4.1** (OpenAI, 2025), and (ii) open-source MLLMs, including **Llama-3.2-11B-Vision** (Dubey et al., 2024) and **Qwen-2.5-VL** (Bai et al., 2025). These models are serve as valuable references to contextualize HydroGen's performance.

## 4.5 RESULTS

Table 2 demonstrates that HydroGen attains superior performance across all semantic and syntactic criteria compared to both proprietary MLLMs and smaller open-source models on the four-year test set. HydroGen achieves a BERT-F1 score of 84.56, surpassing Gemini-2.5-Pro by 12.75 and GPT-4.1 by 14.50. HydroGen enhances Sentence Similarity by 18.15 compared to Gemini-2.5-Pro and by 31.51 relative to Gemma-3. In syntactic evaluation, HydroGen achieves a BLEU score of 62.01 and a ROUGE-L score of 67.78, surpassing GPT-4.1 by 44.75 and 46.21, respectively. In comparison to Typhoon2-Vision, HydroGen shows a substantial increase of 46.96 in BERT-F1 and 44.79 in Sentence Similarity. For qualitative illustrations of HydroGen's generated reports, please refer to Appendix A.

Table 3: Ablation study of HydroGen components on semantic and syntactic metrics (%). Each step adds one component on top of the previous: **+CPT** denotes the baseline model with continual pre-training; **+SFT** incorporates supervised fine-tuning in addition to CPT; **+Temporal Prompt** includes temporal prompt tokens; and **Ours (+GuideAC)** combines all preceding components along with GuideAC. Best results for each model are highlighted.

| Model Name | Method | Semantic | | Syntactic | |
|---|---|---|---|---|---|
| | | B-F1↑ | SS↑ | BLEU↑ | R-L↑ |
| | Baseline | 65.93 | 43.10 | 6.40 | 15.72 |
| | + CPT | 63.05 | **78.47** | 13.09 | 22.56 |
| Gemma-3-4B | + SFT | 75.30 | 57.37 | 18.01 | 25.34 |
| | + Temporal Prompt | 79.93 | 75.22 | 23.88 | 31.63 |
| | Ours (+ GuideAC) | **80.54** | 74.65 | **27.19** | **33.89** |
| | Baseline | 33.87 | 28.50 | 0.00 | 4.41 |
| | + CPT | 74.43 | 76.30 | 26.59 | 32.66 |
| Typhoon2-Vision-7B | + SFT | 75.78 | 74.95 | 32.56 | 33.81 |
| | + Temporal Prompt | 82.75 | 77.16 | **32.68** | 37.51 |
| | Ours (+ GuideAC) | **84.54** | **82.47** | 32.63 | **38.39** |

These results confirm that HydroGen, despite having only 7B parameters, surpasses both strong proprietary systems and smaller open-source MLLMs in hydrological report generation. The gains highlight the value of domain-specific knowledge and targeted architectural adaptations. While proprietary models such as Gemini-2.5-Pro and GPT-4.1 are strong general-purpose systems, they lack explicit domain grounding. Smaller open-source models, such as Typhoon2-Qwen2-Vision-7B and Gemma-3-4B, similarly falter in their ability to grasp causal reasoning and temporal dynamics, which are essential for this task. HydroGen addresses these gaps through continual pre-training on hydrological corpora, temporal prompt tokens for sequential image alignment, and GuideAC for causal reasoning. These strategies enable stronger semantic accuracy and syntactic coherence, showing that effective domain alignment can outweigh sheer model size—as seen in HydroGen outperforming larger models such as LLaMA-3.2-11B-Vision.

## 4.6 ABLATION STUDY

We evaluated the contribution of each component within our HydroGen framework, as illustrated in Table 3. Starting from the baseline model. First, we discover that the incorporation of continual pre-training (CPT) markedly enhances semantic similarity and syntactic metrics, including BLEU and ROUGE-L, suggesting that domain-specific pre-training augments both comprehension and fluency of the generated reports. Secondly, supervised fine-tuning (SFT) enhances semantic and syntactic scores, emphasizing the significance of aligning the model with multimodal hydrological datasets.Third, the introduction of temporal prompt tokens results in significant improvements in BLEU and ROUGE-L scores, indicating that explicit chronological indicators enhance the model's ability to generate cohesive narratives.

Lastly, the incorporation of the GuideAC agent results in superior performance across nearly all criteria. This illustrates that causal reasoning via antecedent-consequence principles is essential for producing accurate and well-organized hydrological reports. The results consistently demonstrate that each element—CPT, SFT, temporal prompts, and GuideAC—positively contributes, and their integration produces the most precise and fluent outputs for both the Gemma-3-4B and Typhoon2-Qwen2-Vision-7B models. For a detailed qualitative and quantitative analysis of each component, please refer to Appendix C.

## 5 Conclusion

We introduced HydroGen, a domain-adaptive multimodal framework for hydrological report generation in Thailand that combines (1) typhoon detection with shapefile-based location mapping, (2) continual pre-training on hydrological data, (3) temporal frame inputs, and (4) causal-guided strategies. Evaluated on weekly hydrological reports from 2018–2025, HydroGen substantially outperforms strong multimodal baselines, achieving a BERT-F1 of 84.56% (+46.96%) for semantic accuracy and a ROUGE-L of 67.78% (+61.48%) for syntactic fidelity. These results underscore the value of integrating domain knowledge and temporal reasoning into multimodal models, and highlight HydroGen's potential to advance AI-driven scientific reporting and climate monitoring.

## 6 Limitations and Future Works

Our study has two primary drawbacks. First, the instruction dataset comprises weekly reports from 2018 to October 2025, although extensive in duration, it is constrained in size and dependent on a singular source. These qualities may limit generalizability and emphasize the necessity of integrating supplementary relevant datasets. Second, the framework is tailored for Thailand, using shapefile mapping specific to its provinces. This leaves it less adaptable to other regions without the incorporation of varied geographical shapefiles and hydrological datasets.

For future work, we intend to extend HydroGen beyond Thailand by adapting it to other regions, requiring integration with diverse geographical shapefiles and regional hydrological datasets. We also aim to explore emerging vision reasoning models that can provide stronger temporal and causal understanding of meteorological imagery. In addition, we plan to incorporate expert-involved evaluations, including blind rating of generated reports and expert validation of GuideAC's causal rules, to further assess factual reliability and improve the robustness of our rule-based reasoning component.

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

## A  HYDROGEN RESULTS

### A.1  HYDROGEN RESULTS ON NORMAL EVENTS

The input and output example of HydroGen for a normal event is displayed in Figure 2.

**28 April 2025 - 4 May 2025**

**Ground Truth**

This week, **a high-pressure system** from **China extended over the South China Sea, the North, and the Northeast.** There was a convergence of southeasterly and southwesterly winds over the North and upper Northeast during the first half of the week, with a westerly trough from Myanmar moving into the area afterward. **Southerly and southeasterly winds then covered upper Thailand,** resulting in hot weather and scattered light to **moderate rainfall in many areas, with heavy rainfall in some areas** throughout the week. **The South** was covered by northeasterly winds over the Andaman Sea and the Gulf of Thailand during the first half of the week, and by northwesterly winds during the latter half of the week, resulting in **rainfall** throughout the week.

**HydroGen**

Over the past week, Thailand experienced dense cloud cover in many areas, particularly in the North, Northeast, Central, and East. This was due to **a high-pressure system** extending over **upper Thailand**, combined with **southeasterly and southerly winds blowing across the upper part of the country.** This resulted in **moderate to heavy rainfall in some areas of upper Thailand.** Meanwhile, the South had light cloud cover during the first half of the week and dense cloud cover during the second half, caused by easterly winds blowing across the South and the Gulf of Thailand, leading to **moderate to heavy rainfall** in some areas of **the South**.

Figure 2: Qualitative result example of HydroGen. The correct elements compared to the ground truth from experts are highlighted in blue.

### A.2  HYDROGEN RESULTS ON EXTREME EVENTS

Figure 3 shows a representative extreme-event week in mid–July 2025, during which **Tropical Storm Wipha** developed and affected regions across Thailand. In contrast, Figure 4 presents a late–September 2025—**Typhoon Ragasa** and **Tropical Storm Bualoi**—resulting in overlapping storm activity across Southeast Asia.

In both extreme-events showing that HydroGen can capture key atmospheric patterns, monsoon-trough movements, and rainfall impacts with clear and coherent explanations. Although the model yet fails to identify official typhoon names in Figure 3—a limitation also noted in our error analysis (Appendix C.3.3)—it still describes cloud development, pressure changes, and resulting weather conditions accurately. These results suggest that HydroGen remains robust not only in normal situations but also when dealing with intense and complex multi-storm events.

## B  HYDROGEN PROMPTS

To guide report generation, HydroGen relies on carefully designed prompting strategies that integrate hydrological knowledge, logical writing structures, and causal reasoning rules. We describe the three key prompt types below.

**14 July 2025 - 20 July 2025**

**Ground Truth**

This week, the upper part of Thailand is covered by **dense clusters of clouds across the country** throughout the week, influenced by the **monsoon trough passing over the upper northern region of Thailand, upper Laos**, and northern Vietnam. At the beginning of the week, there were also two low-pressure areas affecting Thailand's weather. The first covered northern Vietnam for almost the entire week, and the second covered upper Laos on **July 16-17, 2025**. Additionally, **Tropical Storm "Wipha,"** which is a strong tropical storm, made landfall in the southern part of Guangdong Province, China, on July 20, 2025, at 7:00 PM. This strengthens the **southwest monsoon covering the Andaman Sea, Thailand, and the Gulf of Thailand**. **This results in continuous rainfall in upper Thailand, especially in the northern, eastern, and upper northeastern regions. Heavy rainfall is expected in some areas on the western side of southern Thailand.**

**HydroGen**

This week, **dense cloud cover is expected over the northern, northeastern, central, and eastern regions of Thailand from July 16-17, 2025**. This is due to the strengthening of the **southwest monsoon over the Andaman Sea, Thailand, and the Gulf of Thailand**. **Additionally, the monsoon trough, which passed over the upper northern and northeastern regions of Thailand and Laos** at the beginning of the week, shifted southward to pass over the lower northern, northeastern, and upper central regions from July 18-20, 2025. **This weather pattern will result in heavy rainfall in some areas of northern Thailand, particularly in the northern, northeastern, central, and eastern regions. The western coast of southern Thailand will also experience heavy rainfall in some areas.**

Figure 3: Qualitative result example of HydroGen. The correct elements compared to the ground truth from experts are highlighted in blue.

### B.1 STYLE TRANSFER PROMPT

This prompt leverages validated logic and few-shot examples written by hydrological experts in a cause–effect writing style (CURATED_EXAMPLES). The aim is to transfer expert reasoning patterns into the model's outputs. It enforces constraints such as always including cloud descriptions before mentioning rainfall, maintaining chronological order, and varying sentence openers.

**Example Prompt:**

> **Style Transfer Prompt**
>
> Please rewrite the 'input context' into a single paragraph using a "Cause & Effect" writing style (please keep the same context as before changing the style), based on the following requirements:
> - Must be in Thai language only.
> - The paragraph must follow this logical sequence: Cloud description from satellite images → Weather map/atmospheric explanation from air pressure map → Rainfall or weather effects.
>
> **Context:** {CONTEXT}
>
> **Example:** {CURATED_EXAMPLES}
>
> **Answer:**

Figure 4: Qualitative result example of HydroGen. The correct elements compared to the ground truth from experts are highlighted in blue.

## B.2   REPORT GENERATION PROMPT

This prompt applies GuideAC rules, hydrological knowledge, and report metadata to produce a final weekly report. It integrates both textual metadata and multimodal signals (satellite images and air pressure maps). The model is instructed to generate outputs in Thai only, following temporal constraints that match weekly reporting periods.

**Example Prompt:**

---

**Report Generation Prompt**

**Instruction:** Generate an analytical report on hydrological and disaster-related events for each region of Thailand and its neighboring countries, based on the provided air pressure and satellite images. Please respond in Thai only.

**Metadata:** {METADATA}

**Context:** The report will be issued on date {DATE} for document {DOC_NAME} (there are a total of {NUM_IMAGES} images arranged in order).

**Answer:**

---

## B.3 ANTECEDENT–CONSEQUENCE PROMPT FOR GUIDEAC

The GuideAC prompt is designed to extract causal rules in the form of antecedent–consequence relationships. The output is represented as JSON triples, forming the basis of a hydrology-specific knowledge graph. Nodes represent causes, cause areas, effects, and effect areas, while edges are labeled `cause`. This ensures clear mapping between atmospheric conditions, affected regions, and resulting outcomes.

**Example Prompt:**

---

**GuideAC Prompt**

You are given hydrological text and must extract explicit *cause–effect–area* relations.

1. Extract only direct cause–effect relations (no assumptions).
2. Nodes are labeled as `CAUSE`, `CAUSE_AREA`, `EFFECT`, `EFFECT_AREA`.
3. Relations use the format: `<cause> --> <cause_area> --> <effect> --> <effect_area>`.
4. Output only in JSON format.

**Context:** {CONTEXT}

**Answer:**

---

## C ANALYSIS

Our investigation focuses on three different aspects of HydroGen. First, we study the impact of domain adaptation by continual pre-training, demonstrating that although it boosts quantitative measurements, it may also lead to repetition and restricted lexical diversity in reports. Secondly, we examine causal reasoning through GuideAC, where the sliding windows technique for rule aggregation yields superior performance compared to applying an all-time context, causing more accurate and meaningful outputs. Third, we assess the typhoon detection module that uses YOLOv8-m, which shows strong performance compared to specialized baselines and effectively recognizes clouds and typhoons in satellite imagery.

### C.1 IMPACT OF DOMAIN ADAPTATION

Table 3 outlines the results of an ablation study regarding the components of HydroGen, encompassing the impact of Continual Pre-Training (CPT). The quantitative metrics show that adding CPT leads to a substantial improvement in model performance, with large jumps in both semantic and syntactic scores. This demonstrates that CPT helps the model better capture hydrological patterns in terms of measurable metrics.

Nonetheless, the qualitative findings, as depicted in Figure 5, demonstrate that CPT alone fails to deliver high-quality reports. The obtained results typically show redundancy or are derived directly from the hydrological documents utilized during the CPT. Since CPT optimizes the model mostly for next-token prediction in hydrology text, rather than for report-generating tasks. As a result, while CPT improves numerical performance, it does not guarantee coherent or contextually rich reports, which points out the importance of task-specific fine-tuning or guidance strategies.

Furthermore, when examining a different set of qualitative samples, another limitation emerges. As shown in Figure 6, the CPT-enhanced model consistently begins reports with the same fixed starting phrase across different inputs. This lack of lexical diversity indicates that the model memorizes frequent surface-level patterns from the pre-training corpus instead of adapting flexibly to different reporting contexts. Such starting phrase bias reduces the naturalness of generated reports and further demonstrates that CPT alone is insufficient for producing robust hydrological narratives without additional constraints or prompt guidance.

Figure 5: Qualitative examples of HydroGen outputs after continual pre-training, showing repetitive content as highlighted in red despite metric improvements.

## C.2 CAUSAL REASONING WITH GUIDEAC

The aggregation of antecedent-consequence (AC) rules in HydroGen is essential for efficient causal reasoning. Our results demonstrate that the optimal approach for AC rule aggregation is the *sliding windows* strategy. This method constrains the context for report generation to a defined temporal window, rather than utilizing all previous data, hence limiting the risk of incorporating irrelevant or misleading information. For instance, while producing a report for May 2025, the AC rules context is confined to April and May, eliminating the report date to avert information leakage. The context encompasses data from the current and preceding year, offering adequate historical knowledge without incorporating future data.

Table 4 illustrates that employing sliding windows surpasses the utilization of all-time context when assessed with our fine-tuned Typhoon2-Vision-7B, indicating that temporally limited context enhances the precision and pertinence of the generated reports.

**GuideAC Sensitivity and Robustness.** We additionally evaluate the robustness of GuideAC by altering the rule-set size (Top–15 against Top–10) and the temporal window duration (2-month versus 3-month). Table 5 illustrates that performance is constant across all configurations: a reduced rule set (Top–10) exhibits marginally higher efficiency compared to Top–15, and shorter temporal

Figure 6: Qualitative examples of HydroGen outputs after continual pre-training, showing identical starting phrases across different samples as highlighted in purple.

Table 4: Performance comparison of AC rule selection methods. Sliding windows outperform all-time context for MLLMs.

| Model Name | Method | Semantic | | Syntactic | |
|---|---|---|---|---|---|
| | | B-F1↑ | SS↑ | BLEU↑ | R-L↑ |
| Typhoon2-Vision-7B | all-time GuideAC | 82.67 | 76.42 | **34.01** | 36.40 |
| | sliding windows GuideAC | **84.54** | **82.47** | 32.63 | **38.39** |

windows consistently yield better results than longer ones, suggesting that excessive historical context causes noise. Optimal outcomes are attained with Top-10 rules and a two-month timeframe, verifying that GuideAC is robust to configuration variations and derives the greatest advantage from a concise, temporally concentrated rule set.

## C.3 FAILURE CASE ANALYSIS

After conducting a systematic error analysis on hydrological reports from 2025, we identified 14 cases where HydroGen still exhibits notable shortcomings. These errors fall into three primary categories:

Table 5: Sensitivity analysis of GuideAC under different rule aggregation sizes and sliding-window lengths. The recommended configuration (Top–10 rules, 2-month window) achieves the strongest overall performance.

| Configuration | Semantic | | Syntactic | |
|---|---|---|---|---|
| | B-F1↑ | SS↑ | BLEU↑ | R-L↑ |
| Top–15 rules, SW = 2 months | 84.07 | 80.43 | 56.69 | 57.87 |
| Top–10 rules, SW = 3 months | 81.52 | **80.54** | 54.10 | 56.32 |
| Top–10 rules, SW = 2 months (Ours) | **84.56** | 79.52 | **62.01** | **67.78** |

### C.3.1 MILD HALLUCINATION (5 CASES)

These errors correspond to small factual drifts, such as slightly incorrect region descriptions or softened cause–effect interpretations. For example, the model may shift cloud-affected areas (e.g., predicting impacts in the "upper Northeast" instead of the "lower Northeast"). While these halluci-nations do not fundamentally alter the overall hydrological meaning, they reveal a need for stronger spatial grounding. As shown in Figure 7, these mild hallucinations typically involve subtle spa-tial misalignments rather than major narrative distortions. As future work, we plan to incorporate region-specific grounding vectors or soft spatial constraints derived from shapefiles to reduce re-gional drift.

**14 January 2025 - 20 January 2025**

**Ground Truth**

This week, a strong high-pressure system is covering northern Thailand and the South China Sea from the beginning to the middle of the week, weakening toward the end of the week. This results in cold weather in the North and Northeast, while the Central and Eastern regions will experience cool weather throughout the week. For the Northeast monsoon, it is relatively strong, covering the Gulf of Thailand, the South, and the Andaman Sea. This will cause rain in the central South throughout the week, with heavy rain in some areas.

**HydroGen**

This week, dense cloud cover is expected over the northern and northeastern regions from January 14th to 15th, 2025, while the central region will experience dense cloud cover from January 14th to 16th, 2025. The eastern region will have dense cloud cover from January 14th to 15th, 2025, and the southern region will have dense cloud cover from January 14th to 15th, 2025. This is due to the high-pressure system **extending over upper Thailand** from the beginning to the middle of the week, combined with the strengthening of the northeasterly winds over upper Thailand and the Gulf of Thailand, as well as the easterly winds over the southern region and the Gulf of Thailand, from the middle to the end of the week. This weather pattern will result in cooler temperatures and rain in some areas of upper Thailand, while the southern region will experience continuous rain and heavy rainfall in some places.

Figure 7: Qualitative examples of HydroGen outputs exhibiting mild spatial hallucinations, where affected regions are slightly misidentified as highlighted in red.

### C.3.2  Temporal Ordering Errors (4 cases)

This category includes misordered event sequences, incorrect weekly references, or interchanged reasoning between satellite-image timelines and pressure-map timelines. For example, the model may describe low-pressure strengthening as occurring later in the week, even though meteorological observations indicate an earlier onset. As shown in Figure 8, these temporal inconsistencies often manifest as swapped or misplaced event orderings highlighted in red. To improve this behavior, we plan to augment the current temporal prompt mechanism with temporal knowledge embeddings that capture sequential progression, storm lifecycle phases, and pressure evolution trends.

Figure 8: Qualitative examples of HydroGen outputs exhibiting temporal ordering errors, where misaligned event sequences are highlighted in red.

### C.3.3  Typhoon Mentions and Naming Issues (5 cases)

During periods with multiple storms, HydroGen occasionally refers to an unspecified typhoon system without using the official storm name designated for the region. This is not a hallucination of nonexistent storms but rather a failure to map the correct naming metadata. As shown in Figure 9, these errors typically manifest as incorrect or missing typhoon names highlighted in red. As a future improvement, we will enrich the metadata with authoritative storm names from meteorological agencies or allow the model to output a placeholder (e.g., "Typhoon ＿＿＿") for expert review. This prevents inconsistent naming and ensures alignment with operational hydrology practices.

Overall, these findings highlight areas where HydroGen can be further strengthened, particularly in spatial grounding, temporal sequencing, and storm naming consistency.

**30 September 2025 - 6 October 2025**

**Ground Truth**

Thailand experienced dense cloud cover across the entire country at the beginning of the week. After that, the clouds concentrated in some areas during the middle of the week and covered the entire country densely again at the end of the week. Additionally, **Typhoon "MATMO"** over Luzon Island in the Philippines, moving toward the northern South China Sea, strengthened the southwest monsoon winds over the Andaman Sea, the southern region, and the Gulf of Thailand. This resulted in continuous rainfall, with heavy to very heavy showers in some areas intermittently throughout almost the entire week.

**HydroGen**

Dense cloud cover was observed over the northern, northeastern, central, and eastern regions at the beginning of the week. Subsequently, dense cloud cover concentrated over the northern, northeastern, central, and eastern regions during the middle of the week, and returned to cover the entire country densely at the end of the week. This was due to the strengthening of the southwest monsoon over the Andaman Sea, the southern region, and the Gulf of Thailand during the middle of the week, coupled with a low-pressure area in the northern South China Sea moving closer to northern Vietnam during the middle of the week, resulting in continuous rainfall with heavy to very heavy downpours in some areas.

Figure 9: Qualitative examples of HydroGen outputs exhibiting typhoon naming issues, where incorrect or missing storm names are highlighted in red.

## D   TYPHOON DETECTION PERFORMANCE

We trained the YOLOv8-m model to identify clouds and typhoons in satellite imagery. The training employed datasets from the Hydroinformatics Data Center Platform. The model conducted training on the L4 architecture for 20 epochs.

The performance metrics of the trained model are displayed in Table 6. The metrics indicate that the YOLOv8-m model performs reliably for our specific tasks compared to specialized typhoon detection models, such as TGE-YOLO (He et al., 2025), which is based on the YOLOv8n model and integrates advanced modules like TFAM_Concat and GSConv to improve feature fusion and computational efficiency. Despite these improvements, our YOLOv8-m model demonstrates performance that is adequate with our applications.

Table 6: Comparison of typhoon detection performance on mAP50 (%) between different models.

| Model | mAP50 (%)↑ |
|---|---|
| YOLOv8-n (He et al., 2025) | 79.1 |
| TGE-YOLO (He et al., 2025) | 87.8 |
| **Ours (YOLOv8-m)** | **83.7** |

