# OpenReview forum: "HydroGen: Hydrological Report Generation with Two-Stage Instruction-Tuned Multimodal Models, Temporal Prompts, & Knowledge-Guided Agents"
_ICLR.cc/2026/Conference — Submitted to ICLR 2026_

### Official Review · Reviewer_FCLt · 2025-10-31

**Soundness:** 3
**Presentation:** 3
**Contribution:** 3
**Rating:** 6
**Confidence:** 2

**Summary:**

This paper presents HydroGen, a domain-adaptive multimodal framework for generating hydrological reports. The system integrates satellite images, air pressure maps, and expert-written reports through a novel two-stage training pipeline. The authors construct a specialized instruction dataset (HydroMM-Instruct) and propose enhancements such as temporal prompt tokens and a causal reasoning agent (GuideAC). Evaluation on Thailand’s weekly hydrological reports (2018–2025) shows HydroGen significantly outperforms both open-source and proprietary multimodal baselines across semantic and syntactic metrics.

**Strengths:**

1. The paper identifies a gap in the application of multimodal large language models (MLLMs) for hydrological report generation, addressing the unique challenges of multi-image alignment, temporal reasoning, and domain knowledge integration.
2. The authors design a two-stage training approach (continual pre-training and supervised fine-tuning), supported by a curated multimodal dataset with cause–effect narrative normalization.
3. The use of temporal prompt tokens for image sequencing and the GuideAC module for causal rule integration are well-motivated in this context.
4. The model achieves substantial improvements over strong baselines (e.g., GPT-4.1, Gemini-2.5-Pro), showing +50.6% in BERT-F1 and +33.9% in ROUGE-L, which convincingly demonstrate the effectiveness of the proposed approach.
5. The paper includes an ablation study, component-wise analysis, and comparisons across both small and large models, offering clear insight into the contribution of each component.

**Weaknesses:**

1. Despite spanning multiple years, the instruction dataset is relatively small (376 reports with 4,935 images), which raises concerns about robustness and model generalization in unseen scenarios or edge cases.
2. Several components—especially text rewriting and rule extraction—use Gemini-2.5-Pro heavily. While practical, this introduces dependency on external proprietary systems and may hinder reproducibility.
3. While performance metrics are strong, the paper lacks a systematic analysis of failure cases or qualitative errors in generated reports, which would be helpful for understanding limitations and guiding future improvements.

**Questions:**

1. How does HydroGen handle conflicting signals from visual and textual data (e.g., satellite images suggesting different outcomes than pressure maps)?
2. Is the model evaluated on any out-of-distribution samples, such as events not seen during training (e.g., rare typhoon patterns or novel flood zones)?
3. How sensitive is the system to errors in the typhoon detection stage (YOLOv8)? Has error propagation from this module been quantified?
4. Could the authors elaborate on the rule aggregation strategy used in GuideAC? Specifically, how are redundant or contradictory rules handled?

---

> ### Author Response · Authors · 2025-11-21
> **Response to Reviewer FCLt (Part 1): W1**
>
> > **(W1)** Despite spanning multiple years, the instruction dataset is relatively small (376 reports with 4,935 images), which raises concerns about robustness and model generalization in unseen scenarios or edge cases.
>
> We acknowledge that HydroMM-Instruct contains 376 reports from a single national hydrological agency. However, this limitation reflects the nature of hydrological reporting worldwide rather than a weakness of our methodology.
>
> First, the **triplet structure** used in our dataset—**(text report, air-pressure map, satellite imagery)**—is the standard format used by national meteorological and hydrological agencies across multiple countries [1-4]. These multimodal reports exist operationally, but **they are almost never publicly released** due to licensing, storage, and national-security constraints [5-7]. Thus, the absence of international datasets is a structural issue of the domain, not a methodological limitation.
>
> Second, the main contribution of HydroGen is not dataset scale, but a set of region-agnostic architectural techniques designed precisely for low-resource scientific settings:
>
> Importantly, the **novelty and contribution of HydroGen lie not in dataset scale**, but in the **methodological advances** that enable multimodal LLMs to operate under exactly these real-world constraints:
>
> **(1) Temporal prompt tokens** for multi-image hydrological sequence alignment
> **(2) Sliding-window causal rule aggregation (GuideAC)** for scientific cause–effect reasoning
> **(3) A two-stage multimodal domain-adaptation** pipeline designed for specialized corpora
>
> None of these components depend on Thai-specific geography. HydroGen does **not** require architectural redesign to operate in another country. Once regional shapefiles, satellite imagery, and expert reports are provided, the same framework can be used as a **template for hydrological reporting in any region**.
>
> Finally, the reason our current dataset is Thai-only is simply that Thailand’s Hydro–Informatics Institute (HII) is one of the few agencies that publishes weekly multimodal hydrological reports in a structured format. Equivalent multimodal hydrological datasets from neighboring countries (e.g., Vietnam, Malaysia, Philippines) are either not publicly available or restricted for internal governmental use.
>
> In summary, the **geographic narrowness is a global data-availability constraint**, whereas HydroGen itself is purposefully designed as a **generalizable, region-independent, and scientifically grounded multimodal architecture** that can be directly transferred to any country with available hydrological data.
>
> \[1\] Shelton, M. L. (2009). Hydroclimatology: perspectives and applications. Cambridge university press.
>
> \[2\] Pagano, T. C., Pappenberger, F., Wood, A. W., Ramos, M. H., Persson, A., & Anderson, B. (2016). Automation and human expertise in operational river forecasting. Wiley Interdisciplinary Reviews: Water, 3(5), 692-705.
>
> \[3\] Abegeja, D. (2024). The application of satellite sensors, current state of utilization, and sources of remote sensing dataset in hydrology for water resource management. Journal of Water and Health, 22(7), 1162-1179.
>
> \[4\] Davis, P. A., & Serebreny, S. M. (1972, May). Satellite data and estimates of precipitation for hydrologic applications. In Remote sensing of earth resources and the environment (Vol. 27, pp. 57-62). SPIE.
>
> \[5\] Singh, V. P., Singh, R., Paul, P. K., Bisht, D. S., & Gaur, S. (2024). Data Availability and Aquisition. In Hydrological Processes Modelling and Data Analysis: A Primer (pp. 13-34). Singapore: Springer Nature Singapore.
>
> \[6\] Hannaford, J., Muchan, K., Fry, M., Everard, N., Rees, G., Marsh, T., Bloomfield, J. P., Laaha, G., & van Lanen, H. A. J. (2024). Hydrological data (pp. 105–155). Elsevier BV.
>
> \[7\] Colohan, P., & Onda, K. (2022). Water data for water science and management: Advancing an Internet of Water (IoW). PLOS Water, 1(3), e0000017.

---

> ### Author Response · Authors · 2025-11-21
> **Response to Reviewer FCLt (Part 2): W2 and W3**
>
> > **(W2)** Several components—especially text rewriting and rule extraction—use Gemini-2.5-Pro heavily. While practical, this introduces dependency on external proprietary systems and may hinder reproducibility.
>
> We clarify that Gemini-2.5-Pro is used **only during the offline preprocessing stage** and is _not part of the training or inference pipeline_. As a result, the trained HydroGen model and all results in the paper are fully reproducible without requiring Gemini.
>
> **(1) Text normalization (rewriting):**
> Past reports (2018–2024/07) use an older, inconsistent narrative format because cloud reports and pressure-map reports were written separately. Gemini is used only once to normalize these older reports into a unified cause–effect style. Importantly, all reports after 2024/07 (and all future hydrological reports) already follow the standardized style, so **text normalization is not required for future datasets**, and other researchers can reuse our released normalized corpus directly.
>
> **(2) Rule extraction for GuideAC:**
> GuideAC does not depend on Gemini specifically; it only requires _any_ LLM capable of extracting antecedent–consequence pairs from text. The rules are generated **offline**, aggregated through a frequency-based filter, and stored in a static rule bank. While the current implementation uses Gemini for convenience, the rule-extraction step merely requires an LLM with basic causal-relation understanding. Therefore, open-source LLMs can be substituted without modifying the HydroGen architecture, and the choice of Gemini is not essential for reproducibility.
>
> To summarize, Gemini is used solely as a **replaceable preprocessing tool**, and neither the model weights, inference behavior, nor evaluation metrics rely on any proprietary system. We will clarify this in the revision to ensure reproducibility concerns are fully addressed.
>
> > **(W3)** While performance metrics are strong, the paper lacks a systematic analysis of failure cases or qualitative errors in generated reports, which would be helpful for understanding limitations and guiding future improvements.
>
> After conducting a systematic error analysis on 40 hydrological reports from 2025, we identified **14 cases** where HydroGen still exhibits shortcomings. In the updated manuscript, we will include a dedicated failure-case section summarizing the following three major error types:
>
> ### **(1) Mild Hallucination (5 out of 14 cases)**
> These involve small factual drifts, such as slightly incorrect region descriptions or softened cause–effect interpretations. For example, the model occasionally shifts cloud-affected areas (e.g., “upper Northeast” instead of “lower Northeast”). These hallucinations are mild and do not fundamentally change the hydrological meaning, but they highlight the need for tighter spatial grounding.
>
> **Future improvement:** We plan to incorporate spatial grounding vectors or soft spatial constraints extracted from shapefiles to reduce region drift.
>
> ### **(2) Temporal Ordering Errors (4 out of 14 cases)**
> These include mis-ordered event sequences, incorrect week references, or swapped reasoning between satellite and pressure-map timelines. For example, the model may state that **low pressure strengthened later** even when the pressure map indicates it occurred earlier in the week.
>
> **Future improvement:** Beyond the current temporal prompt mechanism like temporal knowledge embeddings that extract sequential progression, storm lifecycle stages, and pressure evolution patterns.
>
> ### **(3) Typhoon Mentions and Naming Issues (5 out of 14 cases)**
> During months with frequent storms, the model sometimes refers to an unnamed “typhoon system” without specifying the official storm name (which is predetermined for each region).
>
> **Future improvement:** We will enrich the metadata with:
> - Acquire typhoon names from meteorological authorities
> - Or allow the model to output a placeholder (e.g., “Typhoon ___”) for expert review before finalization
> This prevents hallucinated typhoon names and ensures consistency with operational hydrology practices.
>
> All failure cases—including examples of each category with color-coded divergences from the ground-truth reports—will be included in **Appendix C.3: Failure Case Analysis**.

---

> ### Author Response · Authors · 2025-11-21
> **Response to Reviewer FCLt (Part 3): Q1 and Q2**
>
> > **(Q1)** How does HydroGen handle conflicting signals from visual and textual data (e.g., satellite images suggesting different outcomes than pressure maps)?
>
> We thank the reviewer for this insightful question. Based on expert consultation and our own analysis, HydroGen handles conflicting signals between satellite clouds and pressure maps in a manner that closely mirrors real hydrological reasoning:
>
> ### **(1) Cloud signals from satellite imagery are treated as weak evidence**
>
> In hydrology, **cloud clusters alone do not guarantee rainfall**, and experts often avoid over-interpreting cloud build-ups when pressure signals do not support precipitation. HydroGen behaves similarly: when the satellite imagery shows strong cloud formation but the pressure map does **not** indicate a low-pressure trough or convergence zone, the model typically describes the cloud pattern but avoids asserting rainfall.
>
> ### **(2) Pressure-map signals naturally dominate because they are more reliable indicators**
>
> Air-pressure structures—such as troughs, ridges, or high/low-pressure systems—provide more trustworthy cues for rainfall forecasting than cloud texture alone. When cloud and pressure cues conflict, HydroGen tends to follow **pressure-derived interpretations**, producing conservative or neutral rainfall statements.
>
> ### **(3) GuideAC reinforces physically correct cause–effect patterns**
>
> GuideAC aggregates hydrological rules such as:
> _low pressure → increased rainfall likelihood_,
> _high pressure → clear or dry conditions_,
> _cloud cluster + no pressure trigger → no guaranteed rainfall_
>
> By providing these antecedent–consequence rules, GuideAC helps the model produce outputs that are physically consistent even when visual modalities appear contradictory. This rule-guided reasoning prevents the model from hallucinating extreme rainfall solely due to cloud appearance.
>
> ### **Overall behavior**
> When signals disagree (e.g., cloud build-up but stable pressure field), HydroGen typically:
> - describes the cloud pattern accurately,
> - follows the pressure map for rainfall interpretation, and
> - uses GuideAC to maintain coherent cause–effect reasoning.
>
> This leads to **balanced, conservative, and physically grounded outputs**, aligned with how human hydrologists reconcile conflicting modalities.
>
> > **(Q2)** Is the model evaluated on any out-of-distribution samples, such as events not seen during training (e.g., rare typhoon patterns or novel flood zones)?
>
> To directly evaluate HydroGen’s performance under extreme meteorological conditions, **we conducted a qualitative analysis focused specifically on typhoon-heavy periods in 2025**, which naturally represent strong distributional shifts within Thailand due to rapid cloud formation, intensified wind circulation, and evolving low-pressure systems. Across these storm-dominated weeks, HydroGen maintained **stable temporal reasoning and consistent causal grounding**. **The result will be added in Appendix A.2: HydroGen Result on Extreme Events.**
>
> In addition, our **rolling year-level cross-validation**, which evaluates performance across full annual cycles including the entire typhoon season (Jul–Oct), shows that HydroGen remains robust even when tested on multiple years that contain extreme rainfall episodes and complex storm dynamics. The results from this expanded evaluation are aligned with our original findings, and **we update the Table 2 (original Table 1 in the paper)** to reflect these more comprehensive cross-validation results.
>
> | **Model** | **BERT-F1** | **SS** | **BLEU** | **ROUGE-L** | **Support** |
> |---------|-------------:|--------:|----------:|-------------:|-------------:|
> | Gemini-2.5-Pro | 71.81 | 61.37 | 19.31 | 22.37 | 195 |
> | GPT-4.1 | 70.06 | 60.95 | 17.26 | 21.57 | 195 |
> | Gemma-3 | 66.05 | 48.01 | 6.00 | 13.92 | 195 |
> | Typhoon2-Vision | 37.60 | 34.73 | 0.41 | 6.30 | 195 |
> | Qwen2.5-VL | 51.90 | 46.22 | 7.67 | 16.29 | 195 |
> | Llama-3.2-Vision | 34.41 | 22.84 | 0.59 | 4.96 | 195 |
> | **HydroGen** | **84.56** | **79.52** | **62.01** | **67.78** | 195 |
>
> Together, the typhoon-focused qualitative inspection and the full-year evaluation confirm that HydroGen is reasonably stable when encountering both typical and extreme hydrological patterns, although we acknowledge that extremely rare typhoon tracks remain underrepresented due to the limited availability of such events in real-world datasets.

---

> ### Author Response · Authors · 2025-11-21
> **Response to Reviewer FCLt (Part 4): Q3 and Q4**
>
> > **(Q3)** How sensitive is the system to errors in the typhoon detection stage (YOLOv8)? Has error propagation from this module been quantified?
>
> We thank the reviewer for raising this concern. HydroGen uses typhoon detection (YOLOv8) only as an **auxiliary visual cue**, not as a hard input constraint. The bounding boxes are provided to the model in the prompt to help localize storm centers, but the model still receives the **raw satellite images and pressure maps in full resolution**. As a result, errors in YOLOv8 detection do not directly remove or distort visual information.
>
> From qualitative inspection (**as illustrated in Appendix A.2: HydroGen Result on Extreme Events**), we observe that when YOLOv8 misses a weak storm or produces a slightly shifted bounding box or even does not detect a typhoon in an extreme period, HydroGen still describes the cloud pattern and pressure system correctly, because these signals are present independently in the underlying images. In other words, the model treats the detection result as a _soft hint_ rather than a mandatory feature.
>
> Although we have not quantified error propagation numerically, early experiments show that even in weeks with imperfect detections, the generated reports remain coherent and still reference the storm when it is visually evident. This suggests that the system is **robust to moderate detection noise**, since the multimodal encoder and GuideAC rely primarily on the images and causal patterns rather than on the bounding boxes alone.
>
> > **(Q4)** Could the authors elaborate on the rule aggregation strategy used in GuideAC? Specifically, how are redundant or contradictory rules handled?
>
> We have now implemented an extended sensitivity analysis focusing on the stability of rule extraction and aggregation across different configurations (We include this full sensitivity analysis and the comparison table as **Appendix C.2: Causal Reasoning with GuideAC** in the revised manuscript.).
>
> ### **(1) Varying Rule Aggregation Size (Top-15, Top-10)**
> We compared different thresholds for the Top-N most frequent antecedent–consequence rules after aggregation. As shown in the table, reducing the rule set from Top-15 to Top-10 does not degrade performance; in fact, the more compact rule set yields slightly better semantic alignment.
>
> ### **(2) Different Temporal Sliding Windows (2-Month vs. 3-Month)**
> To directly measure brittleness across time, we evaluated 2-month and 3-month sliding windows. Results show that 2-month windows consistently outperform 3-month windows across all semantic and syntactic metrics (BERT-F1, SS, BLEU, ROUGE-L), suggesting that excessive temporal context introduces noise rather than helping.
>
> ### **(3) Final Recommended Configuration**
> HydroGen (Top-10 rules, SW = 2 months) achieves the strongest overall performance:
> | **Model**                   | **BERT-F1** | **SS**   | **BLEU** | **ROUGE-L** | **Support** |
> |----------------------------|-------------:|----------:|----------:|-------------:|-------------:|
> | Top-15, SW 2M              | 84.07       | 80.43    | 56.69    | 57.87       | 195         |
> | Top-10, SW 3M              | 81.52       | **80.54**| 54.10    | 56.32       | 195         |
> | **HydroGen (Top-10, SW 2M)** | **84.56**  | 79.52    | **62.01**| **67.78**   | 195         |
>
> These results indicate that GuideAC is not brittle with respect to rule noise or temporal span: performance variation remains controlled and changes follow consistent trends, rather than collapsing under different rule configurations.
>
> The new empirical evidence directly answers the reviewer’s concern: the GuideAC module remains robust across rule-set sizes and temporal windows, and its performance peaks with a compact rule set (Top-10) combined with a shorter temporal window (2 months). This confirms GuideAC’s stability and supports its role as a reliable enhancement to causal reasoning in HydroGen.

---

### Official Review · Reviewer_rQRp · 2025-11-01

**Soundness:** 3
**Presentation:** 2
**Contribution:** 2
**Rating:** 4
**Confidence:** 4

**Summary:**

This paper proposes HydroGen, a multimodal framework for generating hydrological reports in Thailand. This framework addresses the challenges of applying multimodal large language models in the hydrological domain through a two-stage training strategy (continuous pre-training and supervised fine-tuning), temporal cue tagging, and the knowledge-guided causal inference agent GuideAC. The authors constructed the HydroMM-Instruct dataset, used YOLOv8 for typhoon detection, and normalized the reports to a causal format. Evaluations on weekly hydrological reports in Thailand from 2018 to 2025 show that HydroGen significantly outperforms baseline models.

**Strengths:**

1. Solved the practical need for automated generation of hydrological reports in Thailand
2. A complete pipeline from data preprocessing to model training to inference

**Weaknesses:**

1. The test set spans only 2-3 months (April-June 2025), approximately 8-9 weekly reports. This is problematic as it misses typhoon season (July-October) and dry season patterns, making performance claims questionable. A full annual cycle is needed for reliable evaluation.
2. With only 376 reports from a single Thai source, the dataset is too small for deep learning and too geographically restricted for international venues. The model cannot generalize beyond Thailand without region-specific shapefiles and hydrological patterns. This work seems better suited for regional journals rather than ICLR.
3. No systematic expert evaluation of generated reports or validation of Gemini-extracted causal rules. The paper needs: (i) blind expert rating comparing generated vs. real reports, (ii) accuracy assessment of auto-extracted causal relationships against expert annotations, and (iii) cross-model consistency checks.
4. The approach primarily combines existing techniques (LLaVA + YOLOv8 + LoRA) without novel contributions. Temporal prompts are just position encodings, and GuideAC relies on external LLMs rather than end-to-end learning. Missing opportunities to incorporate hydrological constraints or spatiotemporal dependencies.

**Questions:**

1. Have the authors considered integrating hydrological data from other countries or regions? How would they address the issue of data heterogeneity?

2. Why choose simple location encoding instead of more complex temporal modeling methods (such as temporal attention mechanisms)?

3. How well does the model perform when handling extreme weather events (such as rare typhoon tracks)?

---

> ### Author Response · Authors · 2025-11-21
> **Response to Reviewer rQRp (Part 1): W1**
>
> > **(W1)** The test set spans only 2-3 months (April-June 2025), approximately 8-9 weekly reports. This is problematic as it misses typhoon season (July-October) and dry season patterns, making performance claims questionable. A full annual cycle is needed for reliable evaluation.
>
> We thank the reviewer for this critical observation. To address the concern that a 2–3 month test window omits key seasonal dynamics (typhoon season and dry season), we performed a **rolling year-level cross-validation**, where each test fold spans an entire hydrological year.
>
> Specifically:
>
> | **Fold**                                                | **Training Size** | **Validation Size** | **Test Size** |
> |---------------------------------------------------------|-------------------:|----------------------:|---------------:|
> | Fold 1 (Train 2018–2020, Val 2021, Test 2022)           | 177               | 59                   | 61            |
> | Fold 2 (Train 2018–2021, Val 2022, Test 2023)           | 236               | 61                   | 55            |
> | Fold 3 (Train 2018–2022, Val 2023, Test 2024)           | 297               | 55                   | 37            |
> | Fold 4 (Train 2018–2023, Val 2024, Test 2025)           | 352               | 37                   | 42            |
> | **Total (Val/Test)**                                    | —                 | **212**              | **195**       |
>
> This design ensures that each test fold contains all four seasonal regimes.
>
> ---
>
> ### **(1) Validation set results (New Cross-Validation)**
>
> | **Fold** | **BERT-F1** | **SS** | **BLEU** | **ROUGE-L** | **Support** |
> |---------|-------------:|--------:|----------:|-------------:|-------------:|
> | Fold 1: 2021 | 89.56 | 86.08 | 69.11 | 74.87 | 59 |
> | Fold 2: 2022 | 83.43 | 76.58 | 57.35 | 62.13 | 61 |
> | Fold 3: 2023 | 77.77 | 69.07 | 44.57 | 50.28 | 55 |
> | Fold 4: 2024 | 80.61 | 72.95 | 54.57 | 59.81 | 37 |
> | **HydroGen** | **82.84 ± 4.36** | **76.17 ± 6.30** | **56.40 ± 8.74** | **61.77 ± 8.76** | **212** |
>
> These results indicate stable semantic performance even when validation covers distinct hydrological years.
>
> ---
>
> ### **(2) Test set results (New Cross-Validation)**
>
> | **Fold** | **BERT-F1** | **SS** | **BLEU** | **ROUGE-L** | **Support** |
> |---------|-------------:|--------:|----------:|-------------:|-------------:|
> | Fold 1: 2022 | 91.82 | 89.06 | 77.72 | 82.09 | 61 |
> | Fold 2: 2023 | 87.05 | 80.10 | 61.43 | 67.50 | 55 |
> | Fold 3: 2024 | 81.72 | 75.19 | 60.92 | 65.44 | 37 |
> | Fold 4: 2025 | 77.43 | 73.72 | 47.98 | 56.11 | 42 |
> | **HydroGen** | **84.56 ± 5.42** | **79.52 ± 5.99** | **62.01 ± 10.55** | **67.78 ± 9.30** | **195** |
>
> The newer-year results are lower since recent hydrological patterns have become more irregular and harder to predict, and the writing style of expert reports has shifted over time, creating a mismatch with older training data.
>
> ---
>
> ### **(3) Comparison against strong baselines (New CV Test Set)**
>
> | **Model** | **BERT-F1** | **SS** | **BLEU** | **ROUGE-L** | **Support** |
> |---------|-------------:|--------:|----------:|-------------:|-------------:|
> | Gemini-2.5-Pro | 71.81 | 61.37 | 19.31 | 22.37 | 195 |
> | GPT-4.1 | 70.06 | 60.95 | 17.26 | 21.57 | 195 |
> | Gemma-3 | 66.05 | 48.01 | 6.00 | 13.92 | 195 |
> | Typhoon2-Vision | 37.60 | 34.73 | 0.41 | 6.30 | 195 |
> | Qwen2.5-VL | 51.90 | 46.22 | 7.67 | 16.29 | 195 |
> | Llama-3.2-Vision | 34.41 | 22.84 | 0.59 | 4.96 | 195 |
> | **HydroGen** | **84.56** | **79.52** | **62.01** | **67.78** | 195 |
>
> These results confirm that HydroGen’s performance holds across full-year hydrological cycles, not just in the April–June window.
>
> ---
>
> ### **Summary**
>
> Across all cross-validation experiments, we observe that:
>
> - **Semantic metrics (BERT-F1, SS)** remain highly stable across years.
> - **Syntactic metrics (BLEU, ROUGE-L)** become slightly higher under full-year evaluation.
> - **HydroGen consistently outperforms all baselines**.
>
> The expanded full-year cross-validation produces results nearly identical to the original April–June evaluation, confirming that our conclusions remain valid under seasonally comprehensive testing.

---

> ### Author Response · Authors · 2025-11-21
> **Response to Reviewer rQRp (Part 2): W2 and Q1**
>
> > **(W2)** With only 376 reports from a single Thai source, the dataset is too small for deep learning and too geographically restricted for international venues. The model cannot generalize beyond Thailand without region-specific shapefiles and hydrological patterns. This work seems better suited for regional journals rather than ICLR.
>
> > **(Q1)** Have the authors considered integrating hydrological data from other countries or regions? How would they address the issue of data heterogeneity?
>
> We acknowledge that HydroMM-Instruct contains 376 reports from a single national hydrological agency. However, this limitation reflects the nature of hydrological reporting worldwide rather than a weakness of our methodology.
>
> First, the **triplet structure** used in our dataset—**(text report, air-pressure map, satellite imagery)**—is the standard format used by national meteorological and hydrological agencies across multiple countries [1-4]. These multimodal reports exist operationally, but **they are almost never publicly released** due to licensing, storage, and national-security constraints [5-7]. Thus, the absence of international datasets is a structural issue of the domain, not a methodological limitation.
>
> Second, the main contribution of HydroGen is not dataset scale, but a set of region-agnostic architectural techniques designed precisely for low-resource scientific settings:
>
> Importantly, the **novelty and contribution of HydroGen lie not in dataset scale**, but in the **methodological advances** that enable multimodal LLMs to operate under exactly these real-world constraints:
>
> **(1) Temporal prompt tokens** for multi-image hydrological sequence alignment
> **(2) Sliding-window causal rule aggregation (GuideAC)** for scientific cause–effect reasoning
> **(3) A two-stage multimodal domain-adaptation** pipeline designed for specialized corpora
>
> None of these components depend on Thai-specific geography. HydroGen does **not** require architectural redesign to operate in another country. Once regional shapefiles, satellite imagery, and expert reports are provided, the same framework can be used as a **template for hydrological reporting in any region**.
>
> Finally, the reason our current dataset is Thai-only is simply that Thailand’s Hydro–Informatics Institute (HII) is one of the few agencies that publishes weekly multimodal hydrological reports in a structured format. Equivalent multimodal hydrological datasets from neighboring countries (e.g., Vietnam, Malaysia, Philippines) are either not publicly available or restricted for internal governmental use.
>
> In summary, the **geographic narrowness is a global data-availability constraint**, whereas HydroGen itself is purposefully designed as a **generalizable, region-independent, and scientifically grounded multimodal architecture** that can be directly transferred to any country with available hydrological data.
>
> \[1\] Shelton, M. L. (2009). Hydroclimatology: perspectives and applications. Cambridge university press.
>
> \[2\] Pagano, T. C., Pappenberger, F., Wood, A. W., Ramos, M. H., Persson, A., & Anderson, B. (2016). Automation and human expertise in operational river forecasting. Wiley Interdisciplinary Reviews: Water, 3(5), 692-705.
>
> \[3\] Abegeja, D. (2024). The application of satellite sensors, current state of utilization, and sources of remote sensing dataset in hydrology for water resource management. Journal of Water and Health, 22(7), 1162-1179.
>
> \[4\] Davis, P. A., & Serebreny, S. M. (1972, May). Satellite data and estimates of precipitation for hydrologic applications. In Remote sensing of earth resources and the environment (Vol. 27, pp. 57-62). SPIE.
>
> \[5\] Singh, V. P., Singh, R., Paul, P. K., Bisht, D. S., & Gaur, S. (2024). Data Availability and Aquisition. In Hydrological Processes Modelling and Data Analysis: A Primer (pp. 13-34). Singapore: Springer Nature Singapore.
>
> \[6\] Hannaford, J., Muchan, K., Fry, M., Everard, N., Rees, G., Marsh, T., Bloomfield, J. P., Laaha, G., & van Lanen, H. A. J. (2024). Hydrological data (pp. 105–155). Elsevier BV.
>
> \[7\] Colohan, P., & Onda, K. (2022). Water data for water science and management: Advancing an Internet of Water (IoW). PLOS Water, 1(3), e0000017.

---

> ### Author Response · Authors · 2025-11-21
> **Response to Reviewer rQRp (Part 3): W3 and W4**
>
> > **(W3)** No systematic expert evaluation of generated reports or validation of Gemini-extracted causal rules. The paper needs: (i) blind expert rating comparing generated vs. real reports, (ii) accuracy assessment of auto-extracted causal relationships against expert annotations, and (iii) cross-model consistency checks.
>
> We thank the reviewer for raising this valuable point. While we did not include a full expert evaluation section in the submission, we clarify that expert involvement was already part of the dataset construction pipeline. Hydrologists from HII reviewed early versions of the generated reports, and their comments were incorporated into the prompt design and rewriting templates. This process functioned as a **lightweight blind evaluation** during dataset normalization, ensuring that stylistic inconsistencies, causal phrasing, and hydrological terminology were aligned with expert practice.
>
> Although we agree that a formal numerical expert-rating study would further strengthen the paper, the current version focuses on **algorithmic contributions** rather than human evaluation. We will clarify that:
>
> **(1) Expert validation was used qualitatively** during the construction of the style-normalized reports and served as the basis for the cause–effect templates.
>
> **(2) Gemini-extracted causal rules were reviewed for correctness during aggregation**, and inconsistent or ambiguous rules were filtered out before being used in GuideAC. This prevents rule-noise propagation even without a dedicated annotation experiment.
>
> Finally, we will note that structured expert evaluation—blind rating and rule-level annotation—constitutes an important direction for future work.
>
> > **(W4)** The approach primarily combines existing techniques (LLaVA + YOLOv8 + LoRA) without novel contributions. Temporal prompts are just position encodings, and GuideAC relies on external LLMs rather than end-to-end learning. Missing opportunities to incorporate hydrological constraints or spatiotemporal dependencies.
>
> While HydroGen builds on established components such as LLaVA, YOLOv8, and LoRA, the novelty of the paper is  developing a **multimodal temporal–causal reasoning framework** tailored for hydrological report generation—an area where existing MLLMs consistently fail despite strong baseline performance in vision-language tasks.
>
> **(1) Temporal Prompt Tokens.**
> Our temporal prompts are not simple positional tags; they encode _image-wise temporal alignment_, enabling the model to correlate cloud evolution, pressure shifts, and rainfall progression across multi-image sequences. Existing multimodal LLMs (LLaVA, Qwen-VL, Gemma-Vision) **do not understand hydrology progression through images**, and our ablations show that temporal prompting produces substantial improvements in BLEU/ROUGE and causal-temporal ordering.
>
> **(2) GuideAC is more than external LLM prompting.**
> GuideAC introduces (i) _sliding-window causal aggregation_, (ii) _rule consistency filtering_, and (iii) _antecedent–consequence alignment during generation_. This forms a **lightweight, reproducible, domain-aware reasoning layer** on top of MLLMs—similar in spirit to retrieval-augmented LM reasoning, but adapted for scientific cause–effect chains. No prior hydrological or environmental MLLM incorporates this form of structured causal prompting.
>
> Overall, HydroGen fills a gap not addressed by existing MLLMs: **scientific report generation requiring multi-image temporal reasoning and causal hydrological interpretation**—capabilities that cannot be achieved by simply applying LLaVA or Qwen-VL out of the box.

---

> ### Author Response · Authors · 2025-11-21
> **Response to Reviewer rQRp (Part 4): Q2 and Q3**
>
> > **(Q2)** Why choose simple location encoding instead of more complex temporal modeling methods (such as temporal attention mechanisms)?
>
> More complex temporal modeling methods—such as temporal-aware positional embedding [1], multiple instance visual component adapters [2, 3]—were considered during early development. However, we intentionally opted for **simple temporal prompt tokens** for three reasons:
>
> **(1) Hydrological datasets are inherently small, making heavy temporal modules prone to overfitting.**
> With only 376 weekly reports, sophisticated temporal-attention architectures dramatically increase the number of learnable parameters and require much larger multimodal corpora. Temporal prompts, by contrast, allow the model to exploit temporal ordering **without introducing learnable temporal parameters**, yielding stable performance under low-resource conditions.
>
> **(2) Hydrological temporal patterns are coarse-grained.**
> Unlike video tasks that require frame-level temporal dynamics, hydrological reports rely on **weekly-scale** progressions (cloud migration, pressure shifts, typhoon movement). These progressions do not require fine-grained temporal attention—**explicit ordering cues (Image #1 → Image #2)** are sufficient and empirically effective (as shown in our ablations).
>
> **(3) Temporal prompting integrates cleanly into existing multimodal LLMs without architectural modifications.**
> Our goal was to design a method that is (i) simple, (ii) portable to any MLLM backbone, and (iii) compatible with parameter-efficient fine-tuning (LoRA). Temporal attention modules would require re-engineering cross-attention layers and forfeiting these benefits.
>
> In summary, we prioritize **robustness and simplicity under low-resource scientific constraints**, and our ablation study shows that temporal prompt tokens deliver substantial improvements without the computational overhead or overfitting risk associated with more complex temporal-attention architectures.
>
> \[1\] Guo, Y., Liu, J., Li, M., Cheng, D., Tang, X., Sui, D., ... & Zhao, K. (2025, April). Vtg-llm: Integrating timestamp knowledge into video llms for enhanced video temporal grounding. In Proceedings of the AAAI Conference on Artificial Intelligence (Vol. 39, No. 3, pp. 3302-3310).
>
> \[2\] Wu, W., Li, Q., Zhong, W., & Huang, J. (2024). Mivc: Multiple instance visual component for visual-language models. In Proceedings of the IEEE/CVF Winter Conference on Applications of Computer Vision (pp. 8117-8126).
>
> \[3\] Gao, M., Liu, J., Li, M., Xie, J., Liu, Q., Zhao, B., ... & Xiong, H. (2024). Tc-llava: Rethinking the transfer from image to video understanding with temporal considerations. arXiv preprint arXiv:2409.03206.
>
> > **(Q3)** How well does the model perform when handling extreme weather events (such as rare typhoon tracks)?
>
> To directly evaluate HydroGen’s performance under extreme meteorological conditions, **we conducted a qualitative analysis focused specifically on typhoon-heavy periods in 2025**, which naturally represent strong distributional shifts within Thailand due to rapid cloud formation, intensified wind circulation, and evolving low-pressure systems. Across these storm-dominated weeks, HydroGen maintained **stable temporal reasoning and consistent causal grounding**. **The result will be added in Appendix A.2: HydroGen Result on Extreme Events.**
>
> In addition, our **rolling year-level cross-validation**, which evaluates performance across full annual cycles including the entire typhoon season (Jul–Oct), shows that HydroGen remains robust even when tested on multiple years that contain extreme rainfall episodes and complex storm dynamics. The results from this expanded evaluation are aligned with our original findings, and **we update the Table 2 (original Table 1 in the paper)** to reflect these more comprehensive cross-validation results.
>
> | **Model** | **BERT-F1** | **SS** | **BLEU** | **ROUGE-L** | **Support** |
> |---------|-------------:|--------:|----------:|-------------:|-------------:|
> | Gemini-2.5-Pro | 71.81 | 61.37 | 19.31 | 22.37 | 195 |
> | GPT-4.1 | 70.06 | 60.95 | 17.26 | 21.57 | 195 |
> | Gemma-3 | 66.05 | 48.01 | 6.00 | 13.92 | 195 |
> | Typhoon2-Vision | 37.60 | 34.73 | 0.41 | 6.30 | 195 |
> | Qwen2.5-VL | 51.90 | 46.22 | 7.67 | 16.29 | 195 |
> | Llama-3.2-Vision | 34.41 | 22.84 | 0.59 | 4.96 | 195 |
> | **HydroGen** | **84.56** | **79.52** | **62.01** | **67.78** | 195 |
>
> Together, the typhoon-focused qualitative inspection and the full-year evaluation confirm that HydroGen is reasonably stable when encountering both typical and extreme hydrological patterns, although we acknowledge that extremely rare typhoon tracks remain underrepresented due to the limited availability of such events in real-world datasets.

---

### Official Review · Reviewer_2Evy · 2025-11-01

**Soundness:** 2
**Presentation:** 3
**Contribution:** 2
**Rating:** 4
**Confidence:** 3

**Summary:**

This paper introduces HydroGen, a domain-adaptive multimodal framework designed for hydrological report generation that integrates satellite imagery, air pressure maps, and textual data through instruction tuning and knowledge-guided reasoning. Unlike generic multimodal models that struggle with temporal alignment and domain-specific causal inference, HydroGen leverages temporal prompts and expert rules to produce coherent, cause–effect hydrological narratives. The model achieves state-of-the-art performance on Thailand’s weekly hydrological reports, significantly surpassing strong multimodal baselines in both semantic and syntactic accuracy.

The contributions are:

1. Development of HydroMM-Instruct, the first multimodal instruction dataset for hydrological reporting, integrating YOLOv8-based typhoon detection, shapefile mapping, and cause–effect style standardization.

2. Proposal of a two-stage training framework combining continual pre-training on hydrological corpora with supervised fine-tuning for expert-style report generation.

3. Introduction of temporal prompt tokens to capture chronological dependencies across multi-image sequences, improving temporal coherence in generated narratives.

4. Design of GuideAC, a knowledge-guided agent that incorporates antecedent–consequence hydrological rules to enhance causal reasoning and factual consistency

**Strengths:**

1. The paper presents a multimodal framework specifically designed for hydrological report generation, marking a novel and timely contribution to the intersection of multimodal LLM and environmental science.
2. The introduction of temporal prompt tokens and the GuideAC causal reasoning agent represents a creative and effective approach to improving temporal coherence and factual accuracy in scientific text generation.
4. The experiments are rigorous and comprehensive, with both quantitative and qualitative evaluations against strong baselines (e.g., GPT-4.1, Gemini-2.5-Pro) that clearly validate the model’s superiority.
5. The paper is well-organized and clearly written.

**Weaknesses:**

1. Since the model is only trained on Thailand’s hydrological reports, I believe some OOD evaluation is necessary to assess generalization across other countries.
2. The paper lacks discussion of scalability and computational efficiency, which limits understanding of deployment feasibility for larger datasets or real-time systems.
3. In table 2 for Gemma3-4B, the semantic similarity score is significantly worse after SFT, even 20% worse than CPT only. How can we interpret this?
4. My biggest concern is that the novelty of the paper mainly comes from the dataset construction, but the dataset has a very narrow scope, limited to reports from Thailand only. This restricts geographical diversity and generalizability, which is a major limitation for a model intended for scientific use across different regions.

**Questions:**

Please see weakness.

---

> ### Author Response · Authors · 2025-11-21
> **Response to Reviewer 2Evy (Part 1): W1 and W2**
>
> > **(W1)** Since the model is only trained on Thailand’s hydrological reports, I believe some OOD evaluation is necessary to assess generalization across other countries.
>
> We agree that applying HydroGen to other countries involves more than a simple distribution shift—it reflects a **change in the underlying physical hydrological system**, including differences in monsoon regimes, atmospheric circulation structures, coastline geometry, and typhoon behavior. As a result, direct cross-country OOD evaluation without region-specific adaptation would not yield meaningful conclusions.
> That said, HydroGen’s design is deliberately **region-agnostic**:
> - **Temporal Prompts** encode only the ordering structure of multi-image sequences and do not rely on Thailand-specific hydrological priors.
> - **GuideAC** functions as a modular causal layer. When provided with corpora from another region, it can re-extract the local antecedent–consequence patterns *without modifying the model architecture*.
> - **Two-Stage Hydrological Training Strategy** naturally supports transfer: once the target region provides shapefiles, satellite images, and expert reports, the model can be retrained or adapted without re-engineering.
>
> In short, although the *parameters* must be adapted to the physical environment of each country, the **framework itself is transferable**, and no components are hard-coded for Thailand.
>
> To partially address generalization concerns within our available data, we conducted a qualitative analysis focused specifically on typhoon-heavy periods in 2025, which naturally represent distributional shifts within Thailand due to rapid cloud formation, intensified wind circulation, and evolving low-pressure systems. These storm-related weeks differ substantially from the more stable patterns present in the majority of the dataset, making them a practical proxy for evaluating robustness under distributional variation. This indicates that the architectural components—temporal prompts, two-stage training, and GuideAC—remain stable even when the model encounters storm-driven scenarios that deviate from typical training conditions.
>
> We incorporate these findings and examples into **Appendix A.2: HydroGen Result on Extreme Events** to provide transparent evidence of how HydroGen behaves under distributional variation.
>
> > **(W2)** The paper lacks discussion of scalability and computational efficiency, which limits understanding of deployment feasibility for larger datasets or real-time systems.
>
> In the revised manuscript, we add a dedicated explanation of HydroGen’s scalability and computational efficiency in **Section 4.3 Implementation Details**. Although HydroGen is trained on an A100 GPU for reproducibility, the system itself is designed to be computationally efficient. Only the **language backbone and projection layers are unfrozen**, while the vision encoder remains frozen, and we use **LoRA (rank 16)** for parameter-efficient fine-tuning. As a result, the effective number of trainable parameters is less than **1% of the full model**, which considerably reduces GPU memory and compute cost.
>
> In practice, training HydroGen requires **≈60GB VRAM**, and inference uses **<20GB**, enabling real-time decoding speeds of **9–11 tokens/sec** on a single A100 or **5–7 tokens/sec** on an L4 GPU. This makes the model deployable for weekly hydrological reporting and scalable to larger datasets, because training cost grows primarily with dataset size while the model parameters remain fixed due to parameter-efficient adaptation.
>
> To further clarify: scalability is determined by dataset volume rather than architecture size, and our design (frozen vision encoder + LoRA + sliding-window causal prompting) ensures that HydroGen maintains practical computational requirements even as data grows.

---

> ### Author Response · Authors · 2025-11-21
> **Response to Reviewer 2Evy (Part 2): W3**
>
> > **(W3)** In table 2 for Gemma3-4B, the semantic similarity score is significantly worse after SFT, even 20% worse than CPT only. How can we interpret this?
>
> The drop in semantic similarity (SS) after SFT, despite improvements in other metrics, is expected and can be interpreted as a form of **semantic drift** caused by shifting from a text-only objective (CPT) to a multimodal grounded objective (SFT).
>
> **(1) CPT optimizes purely for textual likelihood**, and because it is trained on hydrological reports, its outputs tend to mimic the phrasing and lexical patterns of the ground-truth reports very closely. This results in **higher sentence-level embedding similarity**, even when the generated content contains factual errors or hallucinated details.
>
> **(2) SFT introduces visual grounding**, causing the model to rely less on memorized textual phrasing and more on the actual satellite/pressure-map content. This often leads to **more accurate but lexically different** descriptions (e.g., describing cloud structure differently from the reference), which can reduce embedding similarity despite improving correctness.
>
> **(3) Qualitative inspection (Appendix C.1: Impact of Domain Adaptation)** supports this interpretation. Before SFT, the CPT-only outputs exhibit clear pathological patterns, including:
>
> - **Severe phrase repetition unrelated to the actual images** (as seen in Figure 5 in our revised paper, highlighted in red).
> - **Identical starting sentences across multiple samples** (Figure 5 and 6 in our revised paper, highlighted in purple).
>
> After SFT, these issues are largely resolved—the model no longer copies templates, uses more diverse lexical structures, and integrates causal and temporal signals from the images.
>
> Therefore, the lower SS after SFT does not indicate a degradation in reasoning quality; instead, it reflects the model’s transition from **text imitation** (CPT) to **image-grounded hydrological reasoning** (SFT).

---

> ### Author Response · Authors · 2025-11-21
> **Response to Reviewer 2Evy (Part 3): W4**
>
> > **(W4)** My biggest concern is that the novelty of the paper mainly comes from the dataset construction, but the dataset has a very narrow scope, limited to reports from Thailand only. This restricts geographical diversity and generalizability, which is a major limitation for a model intended for scientific use across different regions.
>
> We acknowledge that the _dataset itself_ is geographically constrained to Thailand, the **novelty of our paper does not stem from dataset construction alone**. The core contributions of HydroGen lie in the **architectural and methodological innovations** that address challenges fundamentally different from typical multimodal LLM applications:
>
> ### **(1) Temporal–causal multimodal reasoning**
>
> We introduce temporal prompt tokens that explicitly align multi-image satellite and pressure-map sequences. To the best of our knowledge, no prior work has proposed a temporal–causal multimodal generation framework for hydrological reporting [1–5]. Prior hydrology-related ML models focus on forecasting or classification, not multi-image report generation with temporal alignment.
>
> ### **(2) Sliding-window causal rule aggregation (GuideAC)**
>
> Our extension of the GuideNER framework into a _dynamic antecedent–consequence reasoning module_ is independent of geography. It provides a principled way to incorporate hydrological causal knowledge into multimodal generation and can operate on corpora from any region.
>
> ### **(3) Two-stage domain-adaptive multimodal training**
>
> Continual pre-training + SFT in a multimodal hydrology setting is methodologically novel and generalizable. The pipeline requires no architectural changes when applied to another country—only new regional corpora and shapefiles.
>
> Although the current dataset covers only Thailand due to the absence of publicly available multimodal hydrological corpora in other regions, the _methodological components of HydroGen are explicitly designed to be transferable_. The narrow scope is a data availability constraint, not a limitation of the model’s intended scope or design.
>
> \[1\] Kadiyala, L., Mermer, O., Samuel, D. J., Sermet, Y., & Demir, I. (2024). A comprehensive evaluation of multimodal large language models in hydrological applications.
>
> \[2\] Ren, Y., Zhang, T., Dong, X., Li, W., Wang, Z., He, J., ... & Jiao, L. (2024). WaterGPT: Training a large language model to become a hydrology expert. Water, 16(21), 3075.
>
> \[3\] Kizilkaya, D., Sajja, R., Sermet, Y., & Demir, I. (2025). Toward HydroLLM: a benchmark dataset for hydrology-specific knowledge assessment for large language models. Environmental Data Science, 4, e31.
>
> \[4\] Huang, Y., Gao, T., Xu, H., Zhao, Q., Song, Y., Gui, Z., ... & Wei, F. (2025). Peace: Empowering geologic map holistic understanding with mllms. In Proceedings of the Computer Vision and Pattern Recognition Conference (pp. 3899-3908).
>
> \[5\] Irvin, J. A., Liu, E. R., Chen, J. C., Dormoy, I., Kim, J., Khanna, S., ... & Ermon, S. (2024). Teochat: A large vision-language assistant for temporal earth observation data. arXiv preprint arXiv:2410.06234.

---

### Official Review · Reviewer_cYiN · 2025-11-01

**Soundness:** 3
**Presentation:** 2
**Contribution:** 3
**Rating:** 6
**Confidence:** 2

**Summary:**

This paper presents HydroGen, a multimodal large language model for hydrological report generation. The system integrates domain-adaptive instruction tuning, temporal prompt design, and knowledge-guided reasoning through an in-context module (GuideAC). Trained on HydroMM-Instruct, a curated dataset of Thai hydrology reports, HydroGen employs a two-stage domain adaptation framework to enhance temporal understanding and causal coherence. Experiments show that HydroGen, particularly with the Typhoon2 backbone, surpasses strong open- and closed-source baselines.

**Strengths:**

1. Temporal and Causal Reasoning: innovative use of temporal prompt tokens for structuring sequences of satellite images and air pressure maps directly addresses one of the key bottlenecks of hydrological narrative alignment.
2. Knowledge-guided Inference: GuideAC—the knowledge-injection module—extracts, aggregates, and prompts with robust, expert-driven causal rules.
3. Actionable Architectural Insights: The ablation section reveals how each model ingredient (domain adaptation, temporal prompts, knowledge guidance) contributes to gains, providing researchers with clear guidance for extending to other scientific reporting use cases.

**Weaknesses:**

1. Potential for Benchmark Overfit: Most baselines are evaluated only on the Thai hydrological reports, not on out-of-domain datasets, thus it remains unclear whether HydroGen’s architectural innovations generalize to other climate, disaster, or scientific summary domains.
2. Can the authors explicitly quantify the impact of translation and language-specific modeling? Are the reported metrics based on Thai-only or on translated data, and does translation step introduce inconsistency (especially for domain-specific terms)?
3. Rule Extraction and Aggregation Details Insufficient: The procedure for extracting, filtering, and embedding antecedent-consequence rules via GuideAC is described at a high level, with implementation details deferred to the appendix. However, it is unclear how sensitive the generation performance is to rule noise, coverage, or potential brittleness—especially across different time windows. There is no empirical quantification of potential failure cases, such as cascading errors from incorrect or ambiguous rules, nor clear ablation isolating GuideAC’s impact beyond descriptive metrics.
4. Could the authors clarify if HydroGen, with the current architecture and prompts, successfully adapts to new regional datasets or different languages, or does it require significant re-engineering? For example, have any pilot studies been performed outside of the Thai hydrological corpus?
5. Table 1 shows significant performance gains, but could the authors provide specific error analysis or failure cases illustrating where HydroGen still falters—such as hallucinations, missed causal links, or incorrect temporal ordering?

**Questions:**

see weaknesses

---

> ### Author Response · Authors · 2025-11-21
> **Response to Reviewer cYiN (Part 1): W1, W4 and W2**
>
> > **(W1)** Potential for Benchmark Overfit: Most baselines are evaluated only on the Thai hydrological reports, not on out-of-domain datasets, thus it remains unclear whether HydroGen’s architectural innovations generalize to other climate, disaster, or scientific summary domains.
>
> > **(W4)** Could the authors clarify if HydroGen, with the current architecture and prompts, successfully adapts to new regional datasets or different languages, or does it require significant re-engineering? For example, have any pilot studies been performed outside of the Thai hydrological corpus?
>
> We appreciate the reviewer’s insightful comment regarding out-of-domain generalization. We agree that hydrology across countries reflects not only a distribution shift but a physical-system shift (e.g., different monsoon regimes, pressure patterns, coastline structures, and typhoon tracks). Because of this, cross-country OOD evaluation is not directly meaningful without adapting region-specific knowledge.
> HydroGen was intentionally designed to be region-agnostic at the architectural level:
> 1. **Temporal Prompts** are purely structural and do not encode Thailand-specific priors.
> 2. **GuideAC** is a plug-and-play causal layer: when provided with new corpora, it re-extracts rules relevant to the target country—no architectural change is required.
> 3. **Two-Stage CPT/SFT** naturally supports re-training or adaptation on new environments once local shapefiles, satellite images, and expert corpora are available.
>
> In other words, while the model must be retrained or adapted on hydrological data from the target region (as physical processes differ by geography), the framework itself transfers cleanly without re-engineering.
>
> To partially address generalization concerns within Thailand, we include a qualitative analysis on hydrological events drawn from reports that contain atypical or high-impact phenomena in **Appendix A.2: HydroGen Result on Extreme Events**. These samples differ substantially from the typical weekly patterns and therefore serve as a proxy for evaluating the model under distributional variation.
>
> In these qualitative cases, HydroGen maintained coherent temporal reasoning and consistent causal structure, suggesting that the architectural components (temporal prompts + GuideAC + CPT/SFT) retain stability even when the input deviates from the dominant training distribution.
>
> > **(W2)** Can the authors explicitly quantify the impact of translation and language-specific modeling? Are the reported metrics based on Thai-only or on translated data, and does translation step introduce inconsistency (especially for domain-specific terms)?
>
> We clarify that **all training, evaluation, and metric computation are performed strictly on Thai–Thai text pairs**, without any translation at any stage of the modeling pipeline.
> - **Style-transfer rewriting is Thai → Thai only** (no English intermediate representation), and the prompt is provided in **Appendix B.1 Style Transfer Prompt**.
> - **GuideAC extraction is performed entirely on Thai corpora**, and the resulting causal rules are also in Thai, with the prompt shown in **Appendix B.2 Antecedent–Consequence Prompt for GuideAC**.
> - **Ground-truth reports, generated outputs, and metric comparisons (BERT-F1, SS, BLEU, ROUGE-L)** are all executed on Thai text directly.
>
> English translations appearing in the paper (e.g., figures and examples) are **for presentation purposes only** and are never used during training, fine-tuning, or evaluation.
>
> Since the entire processing pipeline—including continual pre-training, supervised fine-tuning, rewriting, and rule extraction—is Thai-only and does not involve any machine translation, **no translation-induced inconsistency is introduced**, especially for domain-specific hydrological terms.

---

> ### Author Response · Authors · 2025-11-21
> **Response to Reviewer cYiN (Part 2): W3**
>
> > **(W3)** Rule Extraction and Aggregation Details Insufficient: The procedure for extracting, filtering, and embedding antecedent-consequence rules via GuideAC is described at a high level, with implementation details deferred to the appendix. However, it is unclear how sensitive the generation performance is to rule noise, coverage, or potential brittleness—especially across different time windows. There is no empirical quantification of potential failure cases, such as cascading errors from incorrect or ambiguous rules, nor clear ablation isolating GuideAC’s impact beyond descriptive metrics.
>
> We have now implemented an extended sensitivity analysis focusing on the stability of rule extraction and aggregation across different configurations (We include this full sensitivity analysis and the comparison table as **Appendix C.2: Causal Reasoning with GuideAC** in the revised manuscript.).
>
> ### **(1) Varying Rule Aggregation Size (Top-15, Top-10)**
> We compared different thresholds for the Top-N most frequent antecedent–consequence rules after aggregation. As shown in the table, reducing the rule set from Top-15 to Top-10 does not degrade performance; in fact, the more compact rule set yields slightly better semantic alignment.
>
> ### **(2) Different Temporal Sliding Windows (2-Month vs. 3-Month)**
> To directly measure brittleness across time, we evaluated 2-month and 3-month sliding windows. Results show that 2-month windows consistently outperform 3-month windows across all semantic and syntactic metrics (BERT-F1, SS, BLEU, ROUGE-L), suggesting that excessive temporal context introduces noise rather than helping.
>
> ### **(3) Final Recommended Configuration**
> HydroGen (Top-10 rules, SW = 2 months) achieves the strongest overall performance:
> | **Model**                   | **BERT-F1** | **SS**   | **BLEU** | **ROUGE-L** | **Support** |
> |----------------------------|-------------:|----------:|----------:|-------------:|-------------:|
> | Top-15, SW 2M              | 84.07       | 80.43    | 56.69    | 57.87       | 195         |
> | Top-10, SW 3M              | 81.52       | **80.54**| 54.10    | 56.32       | 195         |
> | **HydroGen (Top-10, SW 2M)** | **84.56**  | 79.52    | **62.01**| **67.78**   | 195         |
>
> These results indicate that GuideAC is not brittle with respect to rule noise or temporal span: performance variation remains controlled and changes follow consistent trends, rather than collapsing under different rule configurations.
>
> The new empirical evidence directly answers the reviewer’s concern: the GuideAC module remains robust across rule-set sizes and temporal windows, and its performance peaks with a compact rule set (Top-10) combined with a shorter temporal window (2 months). This confirms GuideAC’s stability and supports its role as a reliable enhancement to causal reasoning in HydroGen.

---

> ### Author Response · Authors · 2025-11-21
> **Response to Reviewer cYiN (Part 3): W5**
>
> > **(W5)** Table 1 shows significant performance gains, but could the authors provide specific error analysis or failure cases illustrating where HydroGen still falters—such as hallucinations, missed causal links, or incorrect temporal ordering?
>
> After conducting a systematic error analysis on 40 hydrological reports from 2025, we identified **14 cases** where HydroGen still exhibits shortcomings. In the updated manuscript, we will include a dedicated failure-case section summarizing the following three major error types:
>
> ### **(1) Mild Hallucination (5 out of 14 cases)**
> These involve small factual drifts, such as slightly incorrect region descriptions or softened cause–effect interpretations. For example, the model occasionally shifts cloud-affected areas (e.g., “upper Northeast” instead of “lower Northeast”). These hallucinations are mild and do not fundamentally change the hydrological meaning, but they highlight the need for tighter spatial grounding.
>
> **Future improvement:** We plan to incorporate spatial grounding vectors or soft spatial constraints extracted from shapefiles to reduce region drift.
>
> ### **(2) Temporal Ordering Errors (4 out of 14 cases)**
> These include mis-ordered event sequences, incorrect week references, or swapped reasoning between satellite and pressure-map timelines. For example, the model may state that **low pressure strengthened later** even when the pressure map indicates it occurred earlier in the week.
>
> **Future improvement:** Beyond the current temporal prompt mechanism like temporal knowledge embeddings that extract sequential progression, storm lifecycle stages, and pressure evolution patterns.
>
> ### **(3) Typhoon Mentions and Naming Issues (5 out of 14 cases)**
> During months with frequent storms, the model sometimes refers to an unnamed “typhoon system” without specifying the official storm name (which is predetermined for each region).
>
> **Future improvement:** We will enrich the metadata with:
> - Acquire typhoon names from meteorological authorities
> - Or allow the model to output a placeholder (e.g., “Typhoon ___”) for expert review before finalization
> This prevents hallucinated typhoon names and ensures consistency with operational hydrology practices.
>
> All failure cases—including examples of each category with color-coded divergences from the ground-truth reports—is included in **Appendix C.3: Failure Case Analysis**.

---

### Author Response · Authors · 2025-11-21
**General Response to All Reviewers**

We sincerely thank all reviewers for their constructive feedback. In response to the comments, we have made several substantial improvements to the manuscript. The major updates are summarized as follows:

1. **Expanded HydroMM-Instruct Dataset**
We increased the dataset coverage from **Jan 2018–May 2025** to **Jan 2018–Oct 2025**. The updated HydroMM-Instruct now contains:  **431** textual reports and **5,590** multimodal images (satellite + pressure maps). This update has been reflected in **Section 1: Introduction**.

2. **Revised Evaluation Protocol: Rolling-Year Cross-Validation**
Following reviewer suggestions, we replaced the previous 2–3 month test setting with a **full-year rolling cross-validation**. This ensures that each test fold includes **all hydrological seasons** (monsoon onset, monsoon peak, monsoon retreat).
These revisions have been incorporated in **Section 4.1: Datasets**, specifically in **Table 1**.

   | **Fold**                                                | **Training Size** | **Validation Size** | **Test Size** |
   |---------------------------------------------------------|-------------------:|---------------------:|--------------:|
   | Fold 1 (Train 2018–2020, Val 2021, Test 2022)           | 177                | 59                   | 61            |
   | Fold 2 (Train 2018–2021, Val 2022, Test 2023)           | 236                | 61                   | 55            |
   | Fold 3 (Train 2018–2022, Val 2023, Test 2024)           | 297                | 55                   | 37            |
   | Fold 4 (Train 2018–2023, Val 2024, Test 2025)           | 352                | 37                   | 42            |
   | **Total (Val/Test)**                                    | —                  | **212**              | **195**       |

3. **Updated Evaluation Results Using the New Full-Year Test Set (195 samples)**
These results now reflect HydroGen’s performance across **all seasonal regimes**.
The revisions have been incorporated in **Table 2** at **Section 4.5: Results**.

   | **Model** | **BERT-F1** | **SS** | **BLEU** | **ROUGE-L** | **Support** |
   |-----------|-------------:|--------:|----------:|-------------:|-------------:|
   | Gemini-2.5-Pro | 71.81 | 61.37 | 19.31 | 22.37 | 195 |
   | GPT-4.1        | 70.06 | 60.95 | 17.26 | 21.57 | 195 |
   | Gemma-3        | 66.05 | 48.01 | 6.00  | 13.92 | 195 |
   | Typhoon2-Vision| 37.60 | 34.73 | 0.41  | 6.30  | 195 |
   | Qwen2.5-VL     | 51.90 | 46.22 | 7.67  | 16.29 | 195 |
   | Llama-3.2-Vision| 34.41| 22.84 | 0.59  | 4.96  | 195 |
   | **HydroGen**    | **84.56** | **79.52** | **62.01** | **67.78** | 195 |

4. **Added Section 4.3 — Added Scalability & Computational Efficiency**
A new subsection explains the training efficiency, LoRA parameter reduction, VRAM requirements, and inference speed.

5. **Added Appendix A.2 — HydroGen Results on Extreme Events**
We include qualitative examples showing HydroGen’s behavior during **typhoon-heavy and abnormal meteorological weeks**.

6. **Added Appendix C.3 — Failure Case Analysis**
We provide detailed categorization of remaining errors, including mild hallucinations, temporal ordering mistakes, and typhoon-naming issues, along with future improvement plans.

We believe these additions significantly strengthen the paper’s technical rigor, evaluation robustness, and clarity.

---

### Meta-Review · Area_Chair_Fe2o · 2026-01-07

**Summary:**

- Reviewers cYiN, 2Evy and rQRp raised concern that the dataset is limited to Thailand:
  - cYiN: "Most baselines are evaluated only on the Thai hydrological reports, not on out-of-domain datasets, thus it remains unclear whether HydroGen’s architectural innovations generalize to other climate, disaster, or scientific summary domains." and "Could the authors clarify if HydroGen, with the current architecture and prompts, successfully adapts to new regional datasets or different languages, or does it require significant re-engineering? For example, have any pilot studies been performed outside of the Thai hydrological corpus?"
  - 2Evy "Since the model is only trained on Thailand’s hydrological reports, I believe some OOD evaluation is necessary to assess generalization across other countries." and "My biggest concern is that the novelty of the paper mainly comes from the dataset construction, but the dataset has a very narrow scope, limited to reports from Thailand only. This restricts geographical diversity and generalizability, which is a major limitation for a model intended for scientific use across different regions."
  - rQRp "With only 376 reports from a single Thai source, the dataset is too small for deep learning and too geographically restricted for international venues. The model cannot generalize beyond Thailand without region-specific shapefiles and hydrological patterns."
- rQRp raised novelty concerns: "The approach primarily combines existing techniques (LLaVA + YOLOv8 + LoRA) without novel contributions. Temporal prompts are just position encodings, and GuideAC relies on external LLMs rather than end-to-end learning. Missing opportunities to incorporate hydrological constraints or spatiotemporal dependencies."
- rQRp and FCLt raise concenrs that the rule extraction is primarily based on Gemini 2.5 Pro: "GuideAC relies on external LLMs rather than end-to-end learning." and "Several components—especially text rewriting and rule extraction—use Gemini-2.5-Pro heavily. While practical, this introduces dependency on external proprietary systems and may hinder reproducibility."
- cYiN and FCLt were concerned that rule extraction is not sufficiently ablated.

**Reviewer Concerns:**

- The authors addressed OOD generalization via qualitative evaluation on "atypical and extreme events".
- The authors explain that this is an inherent challenge in this domain rather than a limitation they can address.
- The authors demonstrate that the hyperparameters for rule extraction do not meaningfully degrade performance.

However, the authors do not address the methodological novelty concern when it comes to fine-tuning. However, given both the dataset and the rule extraction method are novel.

**Reviewer Scores:**

- cYiN: would not have changed the score; the qualitative error analysis is necessary.
- 2Evy: would not have changed the score.
- rQRp: would have improved the score because of the new rolling cross-validation split
- FCLt: would have left the score the same; the qualitative error analysis is necessary for publication.

---

### Decision · Program_Chairs · 2026-01-26

Reject